# First-Order Methods for Large-Scale Market Equilibrium Computation

**Yuan Gao**
Department of IEOR, Columbia University
New York, NY, 10027
`gao.yuan@columbia.edu`

**Christian Kroer**
Department of IEOR, Columbia University
New York, NY, 10027
`christian.kroer@columbia.edu`

## Abstract

Market equilibrium is a solution concept with many applications such as digital ad markets, fair division, and resource sharing. For many classes of utility functions, equilibria can be captured by convex programs. We develop simple first-order methods suitable for solving these convex programs for large-scale markets. We focus on three practically-relevant utility classes: linear, quasilinear, and Leontief utilities. Using structural properties of market equilibria under each utility class, we show that the corresponding convex programs can be reformulated as optimization of a structured smooth convex function over a polyhedral set, for which projected gradient achieves linear convergence. To do so, we utilize recent linear convergence results under weakened strong-convexity conditions, and further refine the relevant constants in existing convergence results. Then, we show that proximal gradient (a generalization of projected gradient) with a practical linesearch scheme achieves linear convergence under the *Proximal-PŁ* condition, a recently developed error bound condition for convex composite problems. For quasilinear utilities, we show that Mirror Descent applied to a new convex program achieves sublinear last-iterate convergence and yields a form of Proportional Response dynamics, an elegant, interpretable algorithm for computing market equilibria originally developed for linear utilities. Numerical experiments show that Proportional Response dynamics is highly efficient for computing approximate market equilibria, while projected gradient with linesearch can be much faster when higher-accuracy solutions are needed.

## 1  Introduction

Market equilibrium is a classical model from economics, where the decision of who gets what and why is based on finding a set of market-clearing prices such that every market participant gets allocated an optimal bundle given the prices and their budgets. In this paper, we study the *Fisher market* model, where a set of $n$ buyers compete for $m$ items. In the *competitive equilibrium from equal incomes* (CEEI) mechanism for fair division, a set of $m$ divisible items are to be fairly divided among $n$ agents. The mechanism works by giving every agent a budget of one unit of fake money. Then, a market equilibrium is computed, and the allocation from the equilibrium is used as the fair division of the items, while the prices are discarded since they are for fake money [62]. An allocation given by CEEI can be shown to satisfy various fairness desiderata, e.g., the allocation makes all agents envy-free, Pareto optimal and get at least their proportional share of utility [14, 52]. CEEI has recently been suggested for *fair recommender systems* [42, 43]. Divisible Fisher market equilibrium has also been shown equivalent to the *pacing equilibrium* solution concept for budget smoothing in Internet ad auctions [25, 24]. When items are indivisible, (approximate) CEEI has been used for allocating courses to students [14, 15], and the related max Nash welfare solution for fairly dividing estates and other goods (see, e.g., [17] and also the online fair division service platform

`spliddit.org`). The indivisible setting can be related to the divisible setting via lotteries, though the question of when lotteries can be resolved satisfactorily is intricate [16, 1].

**Market equilibrium.** In this paper we focus on computing Fisher market equilibrium in the divisible setting. Let the market consists of $n$ buyers and $m$ goods. Buyer $i$ has budget $B_i > 0$ and utility function $u_i : \mathbb{R}_+^m \to \mathbb{R}_+$. As mentioned above, $B_i = 1$ for all $i$ corresponds to the CEEI mechanism. Without loss of generality, assume each good $j$ has *unit* supply. An (aggregate) *allocation* is a matrix $x = [x_1^\top; \dots; x_n^\top] \in \mathbb{R}_+^{n \times m}$, where $x_{ij}$ is the amount of item $j$ purchased by buyer $i$. We also refer to $x_i \in \mathbb{R}^m$ as the *bundle* for buyer $i$. Given prices $p \in \mathbb{R}_+^m$ of the goods, the *demand set* of buyer $i$ is defined as the set of utility-maximizing allocations:

$$D_i(p) = \arg\max \left\{ u_i(x_i) \mid x_i \in \mathbb{R}_+^n, \, \langle p, x_i \rangle \leq B_i \right\}.$$

A *competitive equilibrium* is a pair of prices and allocations $(p, x)$, $p \in \mathbb{R}_+^m$ that satisfy the following conditions [32, 39]:

- *Buyer optimality*: $x_i \in D_i(p)$ for all $i \in [n]$,
- *Market clearance*: $\sum_{i=1}^n x_{ij} \leq 1$ for all $j \in [m]$, and equality must hold if $p_j > 0$.

We say that $u_i$ is *homogeneous* (with degree 1) if it satisfies $u_i(\alpha x_i) = \alpha u_i(x_i)$ for any $x_i \geq 0$ and $\alpha \geq 0$ [52, §6.2]. We assume that $u_i$ are concave, continuous, nonnegative, and homogeneous utilities (CCNH). This captures many widely used utilities, such as linear, Leontief, Cobb-Douglas, and general *Constant Elasticity of Substitution* (CES) utilities (see, e.g., [52, §6.1.5] and [10]). For CCNH utilities, a market equilibrium can be computed using the following *Eisenberg-Gale convex program* (EG):

$$\max \sum_{i=1}^n B_i \log u_i(x_i) \quad \text{s.t.} \sum_{i=1}^n x_{ij} \leq 1, \, x \geq 0. \tag{1}$$

More precisely, we state the following theorem, which is well-known in various forms in the literature: see, e.g., [52, §6.2] for the case of differentiable $u_i$ and Eisenberg's original work [31, Theorem 4] for the convex program with a maximization objective of the Nash social welfare). For completeness, we present an elementary, self-contained proof (all proofs are in the Appendix).

**Theorem 1** *For each $i$ assume (i) $u_i$ is finite, concave, continuous and homogeneous on $\mathbb{R}_+^m$; (ii) there exists $x_i \in \mathbb{R}_{++}^m$ (component-wise) such that $u_i(x_i) > 0$. Then, (i) (1) has an optimal solution and (ii) any optimal solution $x^*$ to (1) and the optimal Lagrangian multipliers $p^* \in \mathbb{R}_+^m$ associated with the constraints $\sum_i x_{ij} \leq 1$, $j \in [m]$ form a market equilibrium $(x^*, p^*)$. Furthermore, $\langle p^*, x_i^* \rangle = B_i$ for all $i$.*

By the homogeneity assumption, $u_i(\mathbf{0}) = 0$. The existence of $x_i \in \mathbb{R}_{++}^m$ such that $u_i(x_i) > 0$ is for Slater's condition, a sufficient condition for strong duality. It is satisfied, for example, when $u_i(\mathbf{1}) > 0$, that is, buyer $i$ derives strictly positive utility from all available items. Note that it is possible that $u_i(x_i) < 0$ at some $x_i \geq 0$.

We focus on linear, quasilinear (QL) and Leontief utilities. Under linear utilities, the utility of buyer $i$ is $u_i(x_i) = \langle v_i, x_i \rangle$. As mentioned above, linear utilities are used in the CEEI mechanism in fair division and fair recommender systems. QL utilities have the form $u_i(x_i) = \sum_j (v_{ij} - p_j)x_{ij}$, which depends on the prices [19, 22]. It captures budget-smoothing problems in first- and second-price auction markets [25, 24]. Leontief utilities have the form $u_i(x_i) = \min_{j \in J_i} \frac{x_{ij}}{a_{ij}}$ (see §5). They model perfectly complementary goods and are suitable in resource sharing where an agent's utility is capped by its *dominant resources*, such as allocating different types of compute resources to cloud computing tasks [35, 41].

Another notable convex program that captures market equilibrium under linear utilities is Shmyrev's [59] (see Appendix B.1 and (19) there for more details). [9] shows that the well-known Proportional Response dynamics (PR) [68] for computing market equilibrium under linear utilities is in fact applying Mirror Descent to Shmyrev's convex program (19). In §4, we will show that both Shmyrev's convex program and PR generalize elegantly to QL utilities.

In principle, (1) can be solved using an interior-point method (IPM) that handles exponential cones [60, 26]. However, it does not scale to large markets since IPM requires solving a very large linear

system per iteration. This linear system can often be ill-conditioned and relatively dense, even when the problem itself is sparse. This makes IPM impractical for applications such as fair recommender systems and ad markets, where the numbers of buyers and items are typically extremely large. In this paper, we investigate iterative (gradient-based) first-order methods (FOMs) for computing equilibria of Fisher markets. Because each FOM iteration has low cost, which usually scales only with the number of nonzeros when the problem is sparse, these methods are suitable for scaling up to very large markets. Furthermore, there have been variants of FOMs that allow parallel and distributed updates, making them even more scalable [18, 58, 67, 46]. This is analogous to other equilibrium-computation settings such as zero-sum Nash equilibrium, where iterative first-order methods are also the state-of-the-art [11, 49, 44]. We will focus on the important classes of linear, quasi-linear and Leontief utilities, which are discussed in §3, 4 and 5, respectively. We also note that under CES utilities, PR yields linearly convergent prices and utilities [68, Theorem 4]. Meanwhile, market equilibrium under Cobb-Douglas utilities can be computed explicitly. See Appendix A.6 for more details. We note that there have been highly nontrivial algorithms that find market equilibria in time polynomial in $m, n, \log \frac{1}{\epsilon}$ in theory (where $\epsilon$ is a desired upper bound on error in prices) [27, 63, 7]. However, none of these are as easily implementable as the FOMs we consider here, which also achieves $\epsilon$-equilibrium prices in $O\left(\log \frac{1}{\epsilon}\right)$ time thanks to the linear convergence guarantees. Secondly, these methods all have prohibitively-expensive per-iteration costs from a large-scale perspective.

**First-order methods.**    Consider optimization problems of the form

$$f^* = \min_{x \in \mathcal{X}} f(x) = h(Ax) + \langle q, x \rangle, \tag{2}$$

where $\mathcal{X}$ is a bounded polyhedral set, $h : \mathbb{R}^r \to \mathbb{R}$ is $\mu$-strongly convex with a $L$-Lipschitz continuous gradient on $\mathcal{X}$ (or $(\mu, L)$-s.c. for short), $A \in \mathbb{R}^{d \times r}$ and $q \in \mathbb{R}^d$. We say that an algorithm for (2) converges *linearly* (in objective value) with rate $\rho \in (0, 1)$ if its iterates $x^t$ satisfy $f(x^t) - f^* \leq \rho^t(f(x^0) - f^*)$ for all $t$, where $x^0 \in \text{dom } \mathcal{X}$ is an initial iterate and $f^*$ is the minimum objective value. Unless otherwise stated, $\mathcal{X}^*$ denotes the set of optimal solutions to (2), which is always a bounded polyhedral set [65, Lemma 14].

Various FOMs, such as the following, are naturally suitable for (2).

- Projected gradient (PG): $x^{t+1} = \Pi_{\mathcal{X}}(x^t - \gamma_t \nabla f(x))$, where $\gamma_t$ is the stepsize.
- Frank-Wolfe (FW): $x^{t+1} = x^t + \gamma_t(w^t - x^t)$, where $w^t \in \arg\min_{w \in \mathcal{X}} \langle \nabla f(x^t), w \rangle$ [34, 13, 4, 45]. Unless otherwise stated, $w^t$ is chosen from the set of vertices of $\mathcal{X}$.
- Mirror Descent: $x^{t+1} = \arg\min_{x \in \mathcal{X}} \langle \nabla f(x^t), x - x^t \rangle + \gamma D(x \| x^t)$ [51, 5]. Here, $D$ is the *Bregman divergence* of a differentiable convex function $d$ (see, e.g., [8, §5.3]).

As is well-known, for a broad class of convex optimization problems, these FOMs and their variants achieve *sublinear* convergence, that is, $f(\tilde{x}^t) - f^* = O\left(t^{-k}\right)$ for some $k > 0$, where $\tilde{x}^t$ is either $x^t$ or a weighted average of $x^\tau$, $\tau \leq t$ [13, 8, 3]. When $f$ is strongly convex, linear convergence can be derived. However, strong convexity is highly restrictive and relaxed sufficient conditions for linear convergence of FOMs have been considered. For example, the classical Polyak-Łojasiewicz (PŁ) condition ensures linear convergence of gradient descent [56, 47]. Also have been extensively studied are various *error bound* (EB) conditions, which, roughly speaking, say that the distance from a feasible solution $x$ (possibly required to be close to $\mathcal{X}^*$) to $\mathcal{X}^*$ is bounded by a constant multiple of a computable "residual" of $x$ [48, 61, 65]. Another notable example is the *quadratic growth* (QG) condition, which essentially means that the objective grows at least quadratically in the distance to $\mathcal{X}^*$ [57, 2, 29]. Furthermore, it can be seen to be equivalent to an EB under further assumptions on the problem [29]. Recently, [40] shows that, for convex composite "$f + g$" problems, a so-called *Proximal-PŁ* condition, which generalizes the PŁ condition, is sufficient for linear convergence of proximal gradient. It is also shwon to be equivalent to a few existing conditions under further assumptions on the problem.

**Summary of contributions.** We show that PG, FW and MD are suitable for computing various market equilibria via their convex optimization formulations. In terms of first-order methods, we prove that proximal gradient with a non-standard practical linesearch scheme (Algorithm 1 in the appendix) converges linearly, with a bounded number of backtracking steps per iteration, for the more general class of problems satisfying the Proximal-PŁ condition (Theorem 4). In terms of market equilibria under different utility classes, we establish simple bounds on equilibrium quantities by

exploiting properties of market equilibria (Lemmas 1, 2, 3). Through various problem reformulations exploiting convex optimization duality and using these bounds, we show that the convex programs for ME can be reformulated into ones with highly structured objectives over simple polyhedral sets (2). Then, we derive linear convergence of PG for these convex programs through establishing the Proximal-PŁ condition (Theorems 5,6,8). Specifically, for linear utilities, we show that PG for the EG convex program (1) converges linearly. For QL utilities, based on the relation between EG and Shmrev's convex programs, we derive a new "QL-Shmyrev" convex program (4) whose optimal solutions give equilibrium prices and bids. Similarly, PG for this convex program also achieves linear convergence. We also show that Mirror Descent for the same convex program (4) achieves sublinear last-iterate convergence with a small constant. MD for this convex program also leads to a form of Proportional Response dynamics (PR), a scalable and interpretable algorithm for computing market equilibrium, extending the a series of results for linear utilities [9, 68]. For Leontief utilities, we show that PG for a reformulated dual of (1), with variables being the prices, achieves linear convergence. For all utility classes, linear convergence of running iterates (e.g., prices $p^t$) to their corresponding equilibrium quantities ($p^*$) can be easily derived using linear convergence of the objective values. Extensive numerical experiments demonstrate that PR (and sometimes FW) can quickly compute approximate equilibrium allocations and prices, while PG with linesearch is more efficient for computing a higher-accuracy solution.

**Notation.** The $j$-th unit vector and vector of 1's in $\mathbb{R}^d$ are $\mathbf{e}^{j,(d)}$ and $\mathbf{1}^{(d)}$, respectively, where the superscript $(d)$ is omitted when $d$ is clear form the context. The $d$-dimensional simplex is $\Delta_d = \{x \in \mathbb{R}^d : \mathbf{1}^\top x = 1, \ x \geq 0\}$. For any vector $x$, $x_+$ denotes the vector with entries $\max\{x_i, 0\}$ and $\mathtt{nnz}(x)$ denote its number of nonzeros. All unsubscripted vector norms and matrix norms are Euclidean 2-norms and matrix 2-norms (largest singular value), respectively. For a closed convex set $\mathcal{X} \subseteq \mathbb{R}^d$, $\mathrm{int}(\mathcal{X})$ denotes the interior of $\mathcal{X}$, $\Pi_{\mathcal{X}}(x)$ denotes the Euclidean projection of $x \in \mathbb{R}^d$ onto $\mathcal{X}$ and $\mathrm{Diam}(\mathcal{X}) = \sup_{x,y \in \mathcal{X}} \|x - y\|$. For $p, q \in \mathbb{R}^d_+$, the (generalized) Kullback–Leibler (KL) divergence of $p$ w.r.t. $q$ is $D(p\|q) = \sum_i p_i \log \frac{p_i}{q_i} - \sum_i p_i + \sum_i q_i$. Denote the maximum and minimum *nonzero* singular values of $A$ as $\sigma_{\max}(A)$ and $\sigma_{\min}(A)$, respectively.

## 2 Linear convergence of first-order methods

For PG, it has been shown that linear convergence can be achieved for non-strongly convex objectives under various relaxed conditions [48, 29, 40]. In particular, (2) can be shown to satisfy several such conditions and therefore guarantees linear convergence under PG. One notion of central significance in many such results is the *Hoffman constant*. For $A \in \mathbb{R}^{d \times r}$ and polyhedral set $\mathcal{X} \subseteq \mathbb{R}^d$, denote the (relative) Hoffman constant of $A$ w.r.t. $\mathcal{X}$ as $H_{\mathcal{X}}(A)$, which is the smallest $H > 0$ such that, for any $z \in \mathrm{range}(A)$, $\mathcal{S} = \{x : Ax = z\}$, it holds that $\|x - \Pi_{\mathcal{X} \cap \mathcal{S}}(x)\| \leq H\|Ax - z\|$ for all $x \in \mathcal{X}$ (the Hoffman inequality). Note that $H_{\mathcal{X}}(A)$ is always well-defined, finite, and depends only on $A$ and $\mathcal{X}$ [37, 54]. In the optimization context, the intuition is as follows: the set of optimal solutions $\mathcal{X}^*$ can often be expressed as $\mathcal{X}^* = \mathcal{X} \cap \mathcal{S}$, which means the distance to optimality can be bounded by a constant multiple of the residual $\|Ax - z\|$. Therefore, the Hoffman inequality can be viewed as an EB condition. Further details on this definition, as well as its connection to the classical definition and characterization, can be found in Appendix A.2. Using the Hoffman inequality, we have the following linear convergence guarantee. It is a special case of [40, Theorem 5]. A proof is in Appendix A.3, which is essentially the same as [40, Appendix F]. However, our proof is slightly refined to allow a weakened condition, provide more details on the relevant constants and conclude iterate convergence.

**Theorem 2** *For* (2) *with* $q = 0$*, PG with a constant stepsize* $\gamma_t = \frac{1}{L\|A\|^2}$ *generates* $x^t$ *such that*

$$\frac{\mu}{2H_{\mathcal{X}}(A)}\|x^t - \Pi_{\mathcal{X}^*}(x^t)\|^2 \leq f(x^t) - f^* \leq \left(1 - \frac{\mu}{\max\{\mu, LH_{\mathcal{X}}(A)^2\|A\|^2\}}\right)^t \left(f(x^0) - f^*\right) \text{ for all } t.$$

For the case of $q \neq 0$, we have the following instead, which relies heavily on the boundedness of $\mathcal{X}$. The proof is in Appendix A.4. The key is to make use of the polyhedral structure of the set of optimal solutions and establish a QG condition similar to the one in [4, Lemma 2.3]. Here, we slightly refine the constants using the monotonicity of $f(x^t)$. We note that linear convergence with different rates can also be established through invoking different error bound conditions that also hold for (2) (see, e.g., [65, 29]).

**Theorem 3** *There exists unique $z^* \in \mathbb{R}^r$ and $t^* \in \mathbb{R}$ such that $Ax^* = z^*$ and $\langle q, x^* \rangle = t^*$ for any $x^* \in \mathcal{X}^*$. Furthermore, PG for (2) with constant stepsize $\gamma_t = \frac{1}{L\|A\|^2}$, starting from $x^0 \in \mathcal{X}$, generates $x^t$ such that $\frac{1}{\kappa}\|x^t - \Pi_{\mathcal{X}^*}(x^t)\| \leq f(x^t) - f^* \leq C\rho^t$, where $\rho = 1 - \frac{1}{\max\{1, 2\kappa L\|A\|^2\}}$, $\kappa = H_{\mathcal{X}}(A)^2 \left( C + 2GD_A + \frac{2(G^2+1)}{\mu} \right)$, $C = f(x^0) - f^*$, $G = \|\nabla h(z^*)\|$, $D_A = \sup_{x,x' \in \mathcal{X}} \|A(x - x')\|$.*

For FW, the classical sublinear convergence holds. Specifically, for (2), using FW with static stepsizes or exact linesearch, that is, $\gamma_t = \frac{2}{2+t}$ or $\gamma_t \in \arg\min_{\gamma \in [0,1]} f(x^t + \gamma(w^t - x^t))$, it holds that $f(x^t) - f^* \leq \frac{\text{Diam}(\mathcal{X})^2 L\|A\|^2}{k+2}$ for all $t$ (see, e.g., [38, Theorem 1] and [21, Theorem 2.3]). The advantage of FW is that it can maintain highly sparse iterates when $\mathcal{X}$ has sparse vertices. In fact, if we start with $x^0$ being a vertex, then $x^t$ is a convex combination of $(t+1)$ vertices. For ME convex programs such as (1), the constraint set $\mathcal{X}$ is a product of simplexes $\Delta_n$. Therefore, any vertex of $\mathcal{X}$ has the form $x \in \{0,1\}^{n \times m}$, where each $(x_{1j}, \ldots, x_{nj})$, $i \in [m]$ is a unit vector. Hence, $x^t$ is a linear combination of $(t+1)$ vertices and has at most $(t+1)m$ nonzeros. There have been recent theories on linear convergence of an away-step variant of FW for (2) [4, 45], in which a succinct "vertex representation" [3, §3.2] must be maintained through an additional, potentially costly reduction procedure. Initial trials suggest that it does not work well for our specific convex programs and are therefore not considered further.

**PG with Linesearch.** Linesearch is often essential for numerical efficiency of gradient-based methods, especially when the Lipschitz constant cannot be computed. Here, we consider a modified version of backtracking linesearch for PG (see, e.g., [3, §10.4.2] and [6, §1.4.3]). Specifically, given increment factor $\alpha \geq 1$, decrement factor $\beta \in (0,1)$, and stepsize upper bound $\Gamma > 0$, at each iteration, set the (candidate) stepsize $\gamma_t$ to either the previous one $\gamma_{t-1}$ or $\min\{\Gamma, \alpha\gamma_{t-1}\}$, depending on whether backtracking occurs at all in the previous iteration. Then, perform the usual backtracking with factor $\beta$ until a sufficient decrease condition is met. denote the linesearch subroutine as $(x^{t+1}, \gamma_t, k_t) \leftarrow \mathcal{LS}_{\alpha,\beta,\Gamma}(x^t, \gamma_{t-1}, k_{t-1})$, where $k_{t-1}$ is the number of backtracking steps in the previous iteration, $\gamma_{-1} = \Gamma$, $k_{-1} = 0$. More details, as well as the proof of the following convergence result (in fact, a more general one that holds for the proximal gradient method), can be found in Appendix A.5. Let $L_f^{\mathcal{X}} = \sup_{x,y \in \mathcal{X}, x \neq y} \frac{\|\nabla f(x) - \nabla f(y)\|}{x-y} \leq L\|A\|^2$ be the (effective) Lipschitz constant of $\nabla f$ on $\mathcal{X}$. Note that, in addition to ensure linear convergence, linesearch can potentially improve the constant in the rate, since $L_f^{\mathcal{X}}$ can be much smaller than $L\|A\|^2$.

**Theorem 4** *PG with linesearch $\mathcal{LS}_{\alpha,\beta,\Gamma}$ for (2) with $q = 0$ and $q \neq 0$ converges linearly with rate $1 - \frac{\mu}{\max\{\mu, H^2/\Gamma, H^2 L_f^{\mathcal{X}}/\beta\}}$ (where $H = H_{\mathcal{X}}(A)$) and $1 - \frac{1}{\max\{1, 2\kappa/\Gamma, 2\kappa L_f^{\mathcal{X}}/\beta\}}$, respectively. For both, the number of projections $\Pi_{\mathcal{X}}$ per iteration is at most $1 + \frac{\log \frac{\Gamma}{\tilde{\gamma}}}{\log \frac{1}{\beta}}$, where $\tilde{\gamma} := \min\left\{\Gamma, \beta/L_f^{\mathcal{X}}\right\}$.*

## 3 Linear utilities

Let $u_i(x_i) = \langle v_i, x_i \rangle$ and $v = [v_1, \ldots, v_n] \in \mathbb{R}^{n \times m}$ be the matrix of all buyers' valuations. Without loss of generality, from now on, we assume the following *nondegeneracy* condition, that is, $v$ *does not contain any zero row or column*. Then, the inequality constraints in (1) can be replaced by equalities without affecting any optimal solution. Subsequently, EG with linear utilities can be written as (2) with $\mathcal{X} = (\Delta_n)^m$, $f(x) = h(Ax)$, where $x$ can be viewed as a $(nm)$-dimensional vector, $A \in \mathbb{R}^{n \times nm}$ is a block-diagonal matrix with $i$-th block being $v_i^\top$, $h(u) = \sum_{i=1}^n h_i(u_i)$, $h_i(u_i) := -B_i \log u_i$. Clearly, $\|A\| = \max_i \|v_i\|$. However, in this way, $h$ does not have a Lipschitz continuous gradient on the interior of $\mathcal{U} = \{Ax : x \in \mathcal{X}\}$: for $u \in \text{int}(\mathcal{U})$, each $u_i$ can be arbitrarily close to 0. This can be circumvented by the following bounds on equilibrium utilities.

**Lemma 1** *Let $\underline{u}_i = \frac{B_i\|v_i\|_1}{\|B\|_1}$ and $\bar{u}_i = \|v_i\|_1$, $i \in [n]$. Any feasible allocation $x \in \mathcal{X}$ satisfies $\langle v_i, x_i \rangle \leq \bar{u}_i$ for all $i$. For any equilibrium allocation $x^*$, we have $\langle v_i, x_i^* \rangle \geq \underline{u}_i$ for all $i$.*

Using Lemma 1, we can replace each $h_i$ by its "quadratic extrapolation" when $\langle v_i, x_i \rangle \notin [\underline{u}_i, \bar{u}_i]$. Specifically, let $\tilde{h}(z) := \sum_{i=1}^{n} \tilde{h}_i(u_i)$, where

$$\tilde{h}_i(u_i) := \begin{cases} \frac{1}{2}h_i''(\underline{u}_i)(u_i - \underline{u}_i)^2 + h_i'(\underline{u}_i)(u_i - \underline{u}_i) + h_i(\underline{u}_i) & \text{if } u_i \leq \underline{u}_i, \\ h_i(u_i) = -B_i \log u_i & \text{otherwise.} \end{cases} \tag{3}$$

In the above, clearly, we have $h''(\underline{u}_i) = \frac{B_i}{\underline{u}_i^2} > 0$, $h'(\underline{u}_i) = -\frac{B_i}{\underline{u}_i}$ and $h_i(\underline{u}_i) = -B_i \log \underline{u}_i$. Therefore, $\tilde{h}$ is $\mu$-strongly convex and smooth with $L$-Lipschitz continuous gradient on $\mathcal{U}$, where $\mu = \min_i \frac{B_i}{\|v_i\|_1^2}$ and $L = \max_i \frac{B_i}{\underline{u}_i^2} = \max_i \frac{\|B\|_1^2 \|v_i\|_1}{B_i}$. We can then use the minimization objective $f(x) = \tilde{h}(Ax)$ in (2) instead, without affecting any optimal solution. A direct application of Theorem 3 gives the following convergence guarantee regarding PG for the modified EG convex program. A similar conclusion can also be made using Theorem 4 on PG with linesearch for the same problem.

**Theorem 5** *Let $\mathcal{X}$ and $A$ be as above. Then, PG for the problem $\min_{x \in \mathcal{X}} \tilde{h}(Ax)$ with constant stepsize $\gamma_t = \frac{1}{L\|A\|^2}$ converges linearly with rate $1 - \frac{\mu}{\max\{\mu, LH_{\mathcal{X}}(A)^2\|A\|^2\}}$, where $\|A\| = \max_i \|v_i\|$.*

**Cost per iteration.** In every iteration (or linesearch step), PG requires computing the projection $\Pi_{\mathcal{X}}$, which decomposes item-wise into $\Pi_{\Delta_n}$. This can be computed via an efficient $O(n \log n)$ sorting-based algorithm [23, 20, 66] (in fact, only nonzero elements need to be sorted), while $O(n)$ algorithms are also available [30, 12]. Furthermore, when $v$ is sparse, that is, only a small subset of buyers $I_j \subseteq [n]$ value each item $j$, the total cost for projection scales as $O(\sum_j |I_j| \log |I_j|) = O(\texttt{nnz}(v) \log \texttt{nnz}(v))$, without explicit dependence on $n, m$. Clearly, this is also true for computing the utilities $\langle v_i, x_i \rangle$, $i \in [n]$. Therefore, the cost per iteration scales with number of nonzeros $\texttt{nnz}(v)$ without explicit dependence on $n, m$. Furthermore, the gradient computation $\nabla f(x) = \sum_i \tilde{h}_i'(\langle v_i, x_i \rangle) v_i$ decomposes buyer-wise and $\Pi_{\mathcal{X}}(x) = (\Pi_{\Delta_n}(x_{:,j}))$ decomposes item-wise, allowing straightforward parallelization of PG. In fact, the same holds for subsequent QL and Lenontief utilities.

**Convergent utilities and prices.** The equilibrium utilities $u^*$ are unique [31, Theorem 1]. So are the equilibrium prices $p^*$. Furthermore, $u_i^t = \langle v_i, x_i^t \rangle$ converges linearly to $u^*$ by strong convexity of $h$. In addition, the simplex projection algorithm in each iteration yields a linearly convergent sequence of prices $p^t$ in terms of relative price error $\eta^t := \max_j \frac{|p_j^t - p_j^*|}{p_j^*}$, a commonly used error measure in ME computation [68, 9]. See Appendix B.3 for details.

By Theorem 5 and the above discussion, we have the following, where $u_i^*$ is the (unique) equilibrium utility of buyer $i$. Similar statements can also be made following the subsequent convergence results, i.e., Theorem 6 and 8.

**Corollary 1** *PG for (1) computes an feasible allocation $x^t$ such that $|\langle v_i, x_i^t \rangle - u_i^*| \leq \epsilon$ for all $i$ in $t = O(\tilde{n} \log \tilde{n} \log(1/\epsilon))$ time, where $\tilde{n} = \texttt{nnz}(v)$ and the constant only depends on $v_{ij}$ and $B_i$.*

**Handling additional constraints.** Although efficient computation of $\Pi_{\mathcal{X}}$ depends on the structure of $\mathcal{X}$, it is still arguably more flexible compared to combinatorial algorithms for computing market equilibria, which depend crucially on the specific market structures [64, 63]. Here, for example, the simplex projection algorithms can be modified easily to handle box constraints like $\underline{x}_{ij} \leq x_{ij} \leq \bar{x}_{ij}$ without affecting the time complexity [53, 12]. This allows additional at-most-one constraints $x_{ij} \leq 1$, useful in in fair division applications [42]. Similar bounds on the individual *bids* $b_{ij}$ (which denote how much each buyer spends), such as *spending constraints* (see, e.g., [9, Eq. (2)] and [64]) can also be incorporated without incurring additional cost (this requires solving the Shmyrev convex program (19) which we introduce in Appendix B.1).

## 4   Quasi-linear utilities

In quasi-linear (QL) utilities, the money has value outside the market, which means each buyer's utility is the utility they derive from the items minus their payments, i.e., $u_i(x_i) = \sum_j (v_{ij} - p_j) x_{ij}$, where $p_j$ is the price of item $j$. Note that QL utilities are not CCNH, as it depends on the prices. Nevertheless, based on EG and its dual for linear utilities, another pair of primal and dual convex programs can be derived to capture ME under QL utilities [22, Lemma 5]. In order to derive

Proportional Response dynamics for a market with QL utilities, we now introduce the following convex program, which we call QL-Shmyrev, for its similarity in structure to Shmyrev's convex program for linear utilities (19) [59]. It is the dual of a reformulation of a convex program in [22, Lemma 5]. Let the bids be $b = (b_1, \ldots, b_n)$, where each $b_i = (b_{ij}) \in \mathbb{R}^m$. Denote $p_j(b) = \sum_i b_{ij}$ and $p(b) = (p_1(b), \ldots, p_m(b))$. Introduce slack variables $\delta = (\delta_1, \ldots, \delta_n)$ representing buyers' leftover budgets. Let $\mathcal{B} = \{(b, \delta) \in \mathbb{R}^{n \times m} \times \mathbb{R}^n : (b_i, \delta_i) \in \Delta_{n+1}, \ i \in [n]\}$. The convex program is

$$\varphi^* = \min_{\bar{b}=(b,\delta)} \varphi(b) = -\sum_{i,j}(1 + \log v_{ij})b_{ij} + \sum_j p_j(b) \log p_j(b) \ \text{s.t.} \ \bar{b} \in \mathcal{B}. \tag{4}$$

The derivation is in Appendix C.1. This convex program differs from Shmyrev's convex program (19) for linear utilities in the coefficients of $b_{ij}$ and constraints on $b_{ij}$ (allowing $\sum_j b_{ij} \le B_i$ instead of $=$). For notational brevity, we assume that $v > 0$, although all results hold in the general case of nondegenerate $v$ (no zero row or column) with summations like $\sum_{ij} \log v_{ij}$ being over $(i, j) : v_{ij} > 0$ instead. Here, $\delta_i$ is interpreted as the "leftover" budget of buyer $i$: since $u_i$ depends on the prices $p_j$, a buyer may not spend their entire budget $B_i$, if prices are too high relative to their valuations. In contrast, for CCNH $u_i$, budgets are always depleted at equilibrium. In order to apply PG to solve (4), we need a $(\mu, L)$-s.c. objective, while the function $h(p) = \sum_j p_j \log p_j$ does not have Lipschitz continuous gradient on $\{p(b) : (b, \delta) \in \mathcal{B}\}$, since $p_j(b)$ can be arbitrarily small for $b \in \mathcal{B}$. TO address this, we establish the following bounds on equilibrium prices. We also show that they are unique and equal to the sum of buyers' bids at equilibrium.

**Lemma 2** *For nondegenerate $v$, the equilibrium prices $p^*$ under QL utilities are unique. Let $\underline{p}_j = \max_i \frac{v_{ij} B_i}{\|v_i\|_1 + B_i}$ and $\bar{p}_j = \max_i v_{ij}$ for all $j$. For any optimal solution $(b^*, \delta^*)$ to (4), we have $\underline{p}_j \le p_j^* = p_j(b^*) \le \bar{p}_j$.*

To cast (4) into the standard form (2), we take $\mathcal{X} = \mathcal{B}$, the linear map $A : (b, \delta) \mapsto (p_1(b), \ldots, p_m(b))$, $h(p) = \sum_j p_j \log p_j$ and $q = (q_{ij})$, $q_{ij} = -1 - \log v_{ij}$. Viewing $(b, \delta)$ as a $n(m + 1)$-dimensional vector concatenating each $b_i$ and $\delta$, we have $A := [I, \ldots, I, 0] \in \mathbb{R}^{n \times (n(m+1))}$ and $\|A\| = n$. Then, replace $h$ with a $(\mu, L)$-s.c. function $\tilde{h}$ via a smooth extrapolation similar to that in §3, where $\mu = \frac{1}{\max_j \bar{p}_j}$ and $L = \frac{1}{\min_j \underline{p}_j}$. Let $\tilde{\varphi}(b) = -\sum_{i,j}(1 + \log v_{ij})b_{ij} + \sum_j \tilde{h}(p(b))$. Clearly, $\varphi(b) = \tilde{\varphi}(b)$ as long as $p(b) \in [\underline{p}, \bar{p}]$, since $h = \tilde{h}$ on $[\underline{p}, \bar{p}]$. Combining Theorem 3 and Lemma 2, we have the following.

**Theorem 6** *Let $(b^0, \delta^0)$ satisfies $p(b^0) \in [\underline{p}, \bar{p}]$. Then, PG with stepsize $\gamma_t = \frac{1}{Ln^2}$ for the problem $\min_{(b,\delta)\in\mathcal{B}} \tilde{\varphi}(b)$ converges linearly with rate $1 - \frac{1}{\max\{1, 2\kappa L n^2\}}$, where*
$\kappa = H_{\mathcal{X}}(A)^2\left(C + 2GD + \frac{2(G^2+1)}{\mu}\right)$, $C = \varphi(b^0) - \varphi^*$,
$G = \|\nabla \tilde{h}(p^*)\|$, $D = \sup_{(b,\delta),(b',\delta')\in\mathcal{B}} \|p(b) - p(b')\|$.

**Remark.** Some constants can be bounded explicitly. Clearly, $D \le \sqrt{\sum_j (\max_i v_{ij})^2}$. By Lemma 2, $\tilde{h}(p^*) = h(p^*)$, since $p^* \in [\underline{p}, \bar{p}]$. Therefore, $G = \sqrt{\sum_j (1 + \log p_j^*)^2} \le \sqrt{\sum_j (1 + \log \max_i v_{ij})^2}$.

**MD for** (4) **as Proportional Response dynamics.** Similar to [9], we can also apply MD to (4) and obtain a PR dynamics under QL utilities with $O(1/T)$ convergence in both objective value $\varphi(b^t) - \varphi^*$ and price error $D(p^t\|p^*)$. Recall that MD for (22) with unit stepsize $\gamma_t = 1$ performs the following update:

$$(b^{t+1}, \delta^{t+1}) = \underset{(b,\delta)\in\mathcal{B}}{\arg\min} \langle \nabla\varphi(b^t), b - b^t \rangle + D(b, \delta\|b^t, \delta^t), \tag{5}$$

where $D(b', \delta'\|b, \delta) = \sum_{i,j} b'_{ij} \log \frac{b'_{ij}}{b_{ij}} + \sum_i \delta'_i \log \frac{\delta'_i}{\delta_i}$ is the usual KL divergence. Extending (and slightly strengthening) [9, Eq. (13) and (16)], we first establish the following last-iterate convergence in objective value and price error. Its proof is in Appendix C.3.

**Theorem 7** *Let $b_{ij}^0 = \frac{B_i}{m+1}$, $\delta_i^0 = \frac{B_i}{m+1}$ for all $i, j$. Then, MD applied to (4) generates iterates $(b^t, \delta^t)$, $t = 1, 2, \ldots$ such that $D(p(b^t)\|p^*) \le \varphi(b^t) - \varphi^* \le \frac{\|B\|_1 \log(m+1)}{t}$ for all t.*

The MD update (5) leads to a form of PR dynamics [68, 9] as follows (see Appendix C.4 for the derivation). At time $t$, buyers first submit their bids $b^t = (b_{ij}^t)$. Then, item prices are computed via $p_j^t = \sum_j b_{ij}^t$. Next, each buyer is allocated $x_{ij}^t = b_{ij}^t / p_j^t$ amount of item $j$. Finally, the bids and leftover budgets are updated via

$$b_{ij}^{t+1} = B_i \cdot \frac{v_{ij} x_{ij}^t}{\sum_\ell v_{i\ell} x_{i\ell}^t + \delta_i^t}, \quad \delta_i^{t+1} = B_i \cdot \frac{\delta_i^t}{\sum_\ell v_{i\ell} x_{i\ell}^t + \delta_i^t}. \tag{6}$$

**Convergence of prices and a computable bound.** Similar to the linear case, PG for QL utilities also yields prices $p^t$ with relative error $\eta^t$ converging linearly to 0. Furthermore, it can also be bounded explicitly by computable quantities. See Appendix C.5 for details.

## 5 Leontief utilities

Leontief utilities model perfectly complementary items and are suitable for resource sharing scenarios where utilities are capped by dominant resources. The utility function is $u_i(x_i) = \min_{j \in J_i} \frac{x_{ij}}{a_{ij}}$, where $a_{ij} > 0$ for all $j \in J_i \neq \emptyset$. Denote $I_j = \{i \in [n] : j \in J_i\}$ and assume that $I_j \neq \emptyset$ for all $j$, without loss of generality. Denote $a_i = (a_{i1}, \ldots, a_{im}) \in \mathbb{R}_+^m$, where $a_{ij} := 0$ if $j \notin J_i$ and $a = [a_1^\top; \ldots; a_n^\top] \in \mathbb{R}^{n \times m}$. Under Leontief utilities, EG (1) can be written in terms of the utilities $u \in \mathbb{R}_+^n$ (see (33) in Appendix D.1), whose dual, after reformulation, is

$$\min f(p) = -\sum_i B_i \log\langle a_i, p \rangle \text{ s.t. } p \in \mathcal{P}, \tag{7}$$

where $\mathcal{P} = \{p \in \mathbb{R}_+^m : \sum_j p_j = \|B\|_1\}$. The derivation is in Appendix D.1. In particular, we use the fact that $\|p^*\|_1 = \|B\|_1$ at equilibrium. Similar to the case of linear and QL utilities, we have the following bounds on $\langle a_i, p \rangle$ for Leontief utilities.

**Lemma 3** *For any $p \in \mathcal{P}$, it holds that $\langle a_i, p \rangle \leq \bar{r}_i := \|B\|_1 \|a_i\|_\infty$ for all $i$. Furthermore, for any equilibrium prices $p^*$, it holds that $\langle a_i, p^* \rangle \geq \underline{r}_i := B_i \|a_i\|_\infty$ for all $i$.*

Let $h(u) = -\sum_i B_i \log u_i$. Again, using Lemma 3, we can use a simple quadratic extrapolation to construct a $(\mu, L)$-s.c. function $\tilde{h}$ with $\mu = \min_i \frac{B_i}{\bar{r}_i^2}$, $L = \max_i \frac{B_i}{\underline{r}_i^2}$ without affecting any optimal solution. Clearly, a Lipschitz constant of the gradient of $f(p) = \tilde{h}(ap)$ is $L\|a\|^2$. Analogous to Theorem 5, we have the following convergence guarantee for Leontief utilities. Here, equilibrium prices may not be unique, but the equilibrium utilities $u^*$ are (e.g., by (33) in the appendix). We can also easily construct $u^t$ that converges linearly to $u^*$. See Appendix D.3 for details.

**Theorem 8** *PG with fixed stepsize $\gamma_t = \frac{1}{L\|a\|^2}$ for the problem $\min_{p \in \mathcal{P}} \tilde{h}(ap)$ converges linearly at a rate $1 - \frac{\mu}{\max\{\mu, LH_\mathcal{P}(a)^2\|a\|^2\}}$.*

## 6 Experiments

We perform numerical experiments on market instances under all three utilities with various generated parameters. The algorithms are PGLS (PG with linesearch $\mathcal{LS}_{\alpha,\beta,\Gamma}$, see Appendix A.5), PR and FW (with exact linesearch). For linear utilities, we generate market data $v = (v_{ij})$ where $v_{ij}$ are i.i.d. from standard Gaussian, uniform, exponential, or lognormal distribution. For each of the sizes $n = 50, 100, 150, 200$ (on the horizontal axis) and $m = 2n$, we generate 30 instances with unit budgets $B_i = 1$ and random budgets $B_i = 0.5 + \tilde{B}_i$ (where $\tilde{B}_i$ follows the same distribution as $v_{ij}$). For QL utilities, we repeat the above (same random $v$, same sizes and termination conditions) using budgets $B_i = 5(1 + \tilde{B}_i)$. The termination criterion is either (i) $\epsilon(p^t, p^*) = \max_i \frac{|p_i^t - p_i^*|}{p_i^*} \leq \eta$, where $p^*$ are the prices computed by CVXPY+Mosek [28, 50, 26], or (ii) average duality gap $\mathrm{dgap}_t / n \leq \eta$, for various thresholds values $\eta$. For PGLS, we report the number of linesearch iterations (that is, the total number of projection computations). For other algorithms, we report the number of iterations. As a fair comparison, we use the same parameters $\alpha, \beta, \Gamma$ for PGLS throughout without handpicking. The plots report average numbers of iterations to reach the termination condition and their standard

errors across $k = 30$ repeats. In Appendix E we show: more details on the setup, additional plots with different termination criteria and for Leontief utilities. Codes for the numerical experiments are available at `https://github.com/CoffeeAndConvexity/fom-for-me-codes`.

As can be seen, for linear utilities, PR is more efficient for obtaining an approximate solution (i.e., termination at $\epsilon(p^t, p^*) \leq 10^{-2}$ or $\mathrm{dgap}_t/n \leq 10^{-3}$). When higher accuracy is required, PGLS takes far fewer iterations. For QL utilities, PR is more efficient in most cases, except when very high accuracy is required ($\mathrm{dgap}_t/n \leq 5 \times 10^{-6}$). We do not show FW for the QL case, as it performed extremely badly. In the appendix, we also see that for Leontief utilities, PGLS with linesearch terminates within tens of iterations in all cases.

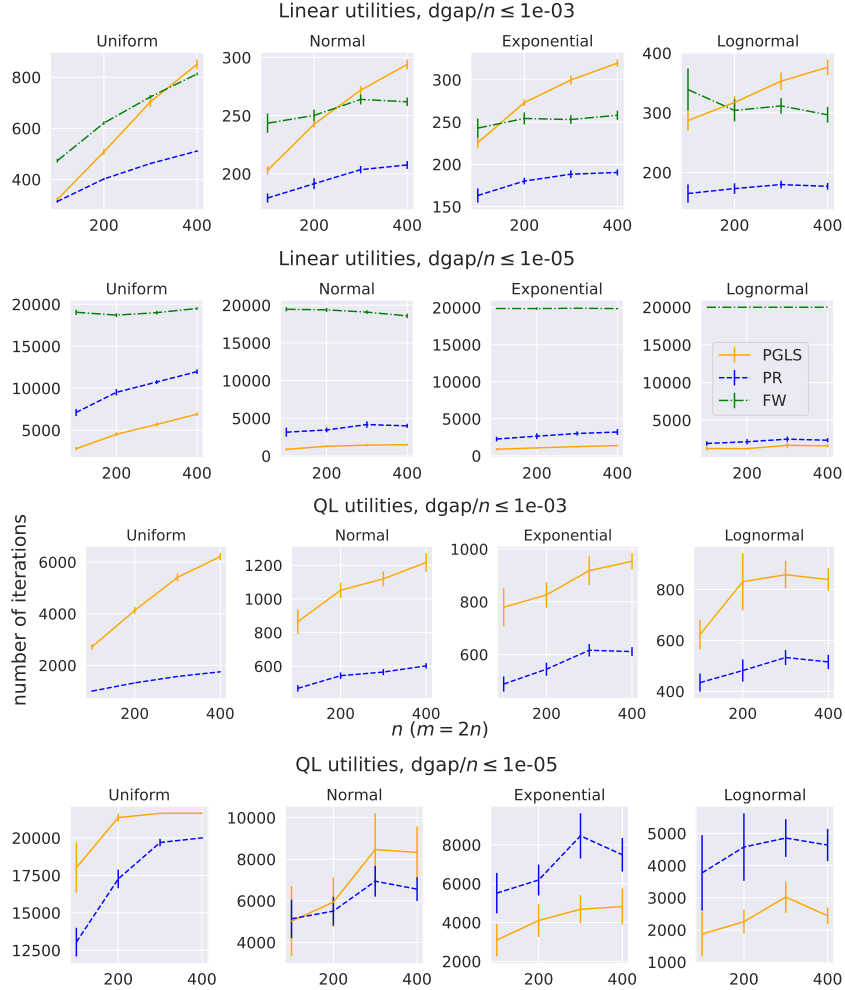

## 7 Conclusions

We investigate the computation of market equilibria under different buyer utility assumptions using simple first-order methods. Through convex optimization duality and properties of market equilibria, we show that the associated convex programs can be reformulated into structured forms suitable for FOMs. We show that projected gradient achieves linear convergence for these reformulations, through weakened strong-convexity conditions. As a technical contribution on FOMs, we also prove a more general linear convergence result of proximal gradient with linesearch under the Proximal-PŁ condition. For QL utilities, we derive a form of Proportional Response through Mirror Descent on a convex optimization formulation, extending the classical convergence results for linear utilities. Finally, extensive numerical experiments compare the efficiency of the FOMs for various low- and high-accuracy termination criteria.

## 8 Broader Impact

As mentioned in the introduction, large-scale market equilibrium computation problems arise in important applications such as Internet advertising markets, course assignment at universities, fair recommender systems and compute resource allocation. As such, this work has the following potential positive societal impact: Based on this work, resource allocation schemes, especially those with desirable fairness properties previously deemed hard to solve at scale (and are thus simplified or disregarded), can be implemented in reasonable time through computing a market equilibrium by a first-order method. Progress of a FOM can be easily monitored via the duality gap while feasibility is guaranteed. Our work also enables greater scalability of certain Internet advertising market equilibrium models. This could be used for greater market efficiency or better monetization of such markets. Whether this is viewed as a positive or negative thing is beyond the scope of our paper.

## 9 Funding Transparency Statement

Over the past 36 months, Yuan Gao was a PhD student at Columbia University and was fully funded by Fu Foundation School of Engineering and Applied Sciences (through Department of Industrial Engineering and Operations Research) throughout the development of the present work.

Over the past 36 months, Christian Kroer was employed full time for a year at Facebook Inc., and was at times employed part-time with Facebook Inc. That employment did not fund the present work.

Part of the numerical experiments were run on the computing server of Columbia University Data Science Institute
https://datascience.columbia.edu/about-us/work-with-us/computing-resources/.

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
