[Supplementary Material 1 · main_camera_ready.pdf]

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

## Footnotes

[1]In fact, the bound $\log(mn)$ in [9, Lemma 13] (which assumes $\|B\|_1 = 1$) can be easily strengthened to $\log m$ via the above derivation. In other words, it does not depend explicitly on the number of buyers (but implicitly through $\|B\|_1$ in general).

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

# A Preliminaries

We list and prove a few elementary lemmas used in subsequent proofs and discussions.

**Lemma 4** *Let $n \geq m$, $A \in \mathbb{R}^{m \times n}$ and $b \in \mathbb{R}^m$ be such that $\mathcal{S} = \{x \in \mathbb{R}^n : Ax = b\} \neq \emptyset$. Let the singular value decomposition (SVD) of $A^\top$ be $A^\top = U\Sigma V^\top$, where $U \in \mathbb{R}^{n \times r}$ and $V \in \mathbb{R}^{m \times r}$ have orthonormal columns, $\Sigma = \mathrm{Diag}(\sigma_1, \ldots, \sigma_r)$, $\sigma_1 \geq \ldots \sigma_r > 0$, $r = \mathrm{rank}(A)$. For any $x \in \mathbb{R}^n$, the projection of $x$ onto $\mathcal{S}$ can be expressed as*

$$\Pi_{\mathcal{S}}(x) = (I - UU^\top)x + U\Sigma^{-1}V^\top b.$$

*In particular, when $A$ has full rank,*

$$\Pi_{\mathcal{S}}(x) = (I - A^\top(AA^\top)^{-1}A)x + A^\top(AA^\top)^{-1}b.$$

*Proof.* The case of full rank $A$ is well-known, see, e.g., [55, Eq. (1)]. For general $A$, $Ax = b \Leftrightarrow V\Sigma U^\top x = b$. Since $V\Sigma$ has full column rank, there exists a unique $w \in \mathbb{R}^r$ such that $V\Sigma w = b$. In fact, $w = \Sigma^{-1}V^\top b$. Therefore,

$$Ax = b \Leftrightarrow U^\top x = \Sigma^{-1}V^\top b.$$

Since $U^\top$ has full row rank, the formula follows directly from the full rank case. $\qquad\square$

**Lemma 5** *Under the same assumptions as Lemma 4, for any $x \in \mathbb{R}^n$, it holds that*

$$\sigma_r \cdot \|x - \Pi_{\mathcal{S}}(x)\| \leq \|Ax - b\|.$$

*Proof.* By Lemma (4) and the fact that $U$ and $V$ have orthonormal columns,

$$\|x - \Pi_{\mathcal{S}}(x)\| = \|U(U^\top x - \Sigma^{-1}V^\top b)\| = \|U^\top x - \Sigma^{-1}V^\top b\|$$
$$= \|\Sigma^{-1}V^\top(Ax - b)\| \leq \|\Sigma^{-1}\|\|Ax - b\|,$$

where $\|\Sigma^{-1}\| = \frac{1}{\sigma_r}$. $\qquad\square$

**Lemma 6** *For $B > 0$ and $g \in \mathbb{R}^d$, let*

$$x^* = \arg\min\left\{\langle g, x\rangle + \sum_{i=1}^d x_i \log x_i : x \geq 0, \mathbf{1}^\top x = B\right\}.$$

*Then, $x_i^* = B \cdot \frac{e^{-g_i}}{\sum_\ell e^{-g_\ell}}$, $i \in [d]$.*

*Proof.* It can be easily verified via KKT optimality conditions. The Lagrangian is

$$L(x, \lambda) = \langle g, x\rangle + \sum_i x_i \log x_i - \lambda(\mathbf{1}^\top x - B).$$

By the first-order condition, for any $\lambda$, the minimizer $x(\lambda)$ of $L(x, \lambda)$ has $x_i(\lambda) = e^{\lambda - 1 - g_i}$. Primal feasibility implies $\sum_i x_i(\lambda) = B \Rightarrow e^\lambda = \frac{B}{\sum_i e^{-(1+g_i)}}$. Therefore,

$$x_i^* = B \cdot \frac{e^{-g_i}}{\sum_\ell e^{-g_\ell}}.$$

$\qquad\square$

## A.1 Proof of Theorem 1

**Existence of an optimal solution.** Since $u_i$ is concave and continuous on $[0, 1]^m$, we have
$$\sup_{x_i \in [0,1]^m} u_i(x_i) = M_i < \infty.$$
For any $x \in \mathcal{X} = \{x \in \mathbb{R}_+^{n \times m} : \sum_i x_{ij} \leq 1, \forall j\}$,
$$F(x) := \sum_i B_i \log u_i(x_i) \leq \sum_i B_i M_i < \infty.$$
For each $i$, since there exists $x_i' > 0$ s.t. $u_i(x_i') > 0$, by homogeneity, $u_i(\alpha x_i') > 0$ for any $\alpha > 0$. Hence, there exists $x \in \text{int}(\mathcal{X})$, i.e., $x > 0$ and $\sum_i x_{ij} < 1$ for all $j$, such that $u_i(x_i) > 0$ for all $i$. Therefore,
$$-\infty < F^* = \sup_{x \in \mathcal{X}} \sum_i B_i \log u_i(x_i) < \infty.$$
Note that the sup is not affected if we change it to $\sup_{x \in \mathcal{X}'}$, where
$$\mathcal{X}' = \{x \in \mathcal{X} : u_i(x_i) \geq \epsilon\}$$
for some sufficiently small $\epsilon > 0$. Here, $\mathcal{X}'$ is a compact set, on which $F$ is continuous. Therefore, $F^* = F(x^*)$ for some $x^* \in \mathcal{X}$.

Next, we show that an optimal solution of (1) gives a ME. Consider the minimization form of (1). Let $p_j \geq 0$ be the dual variable associated with constraint $\sum_i x_{ij} \leq 1$. The Lagrangian is
$$\mathcal{L}(x, p) = \left[ -\sum_i B_i \log u_i(x) + \left\langle p, \sum_i x_i \right\rangle - \sum_j p_j \right].$$
The Lagrangian dual is
$$\max_{p \geq 0} g(p) := \min_{x \geq 0} \mathcal{L}(x, p). \tag{8}$$

As shown above, Since $u_i$ are homogeneous, (1) has a strictly feasible solution with finite objective value. (The dual (8) clearly has a strictly feasible solution $p > 0$ with finite objective value $g(p) = \min_{x \geq 0} \mathcal{L}(x, p)$.) Therefore, strong duality holds by Slater's condition and the KKT conditions are necessary and sufficient for (primal and dual) optimality of a solution pair $(x, p)$ (see, e.g., [8, Appendix D]). Let $x^*$ and $p^*$ be optimal solutions to the primal (1) and dual (8), respectively. Clearly, $u_i(x_i^*) > 0$ for all $i$ (since $F^*$ is finite). By Lagrange duality, we have
$$x^* \in \arg\min_{x \geq 0} \mathcal{L}(x, p^*).$$
In other words, each $x_i^*$ maximizes
$$r_i(x_i, p^*) := B_i \log u_i(x_i) - \langle p^*, x_i \rangle$$
on $x_i \geq 0$. We show that $(x^*, p^*)$ is a market equilibrium.

**Buyer optimality** First, we verify that $\langle p^*, x_i^* \rangle = B_i$ for all $i$. Consider the smooth function
$$\phi(\epsilon) = B_i \log u_i((1 + \epsilon)x_i^*) - \langle p^*, (1 + \epsilon)x_i^* \rangle = r_i(x_i^*, p^*) + B_i \log(1 + \epsilon) - \epsilon \langle p^*, x_i^* \rangle.$$
Note that $\phi$ is differentiable on $(0, 1)$ and
$$\phi'(\epsilon) = \frac{B_i}{1 + \epsilon} - \langle p^*, x_i^* \rangle.$$
Assume that $\langle p^*, x_i^* \rangle < B_i$ for some $i$. Then, $\phi'(\epsilon) > 0$ for $\epsilon \in \left[0, \frac{B_i}{\langle p^*, x_i^* \rangle} - 1\right)$. Hence, replacing $x_i^*$ with $(1 + \epsilon)x_i^*$ for a sufficiently small $\epsilon > 0$ strictly increases $r_i(x_i, p^*)$, contradicting to the choice of $x_i^*$. Hence, $\langle p^*, x_i^* \rangle \leq B_i$. Completely analogously, $\langle p^*, x_i^* \rangle \geq B_i$. Therefore, for each buyer $i$, $x_i^*$ is feasible and depletes its budget $B_i$ under prices $p^*$. Hence, for any $x_i \in \mathbb{R}_+^m$ such that $\langle p^*, x_i \rangle \leq B_i$ (a budget-feasible bundle), since $x_i^*$ maximizes $r_i(x_i, p^*)$, we have
$$B_i \log u_i(x_i^*) - \langle p^*, x_i^* \rangle \geq B_i \log u_i(x_i) - \langle p^*, x_i \rangle.$$
Since $\langle p^*, x_i \rangle \leq B_i = \langle p^*, x_i^* \rangle$, the above implies
$$B_i \log u_i(x_i^*) \geq B_i \log u_i(x_i).$$
Therefore, $u_i(x_i^*) \geq u_i(x_i)$. In other words, $x_i^* \in D_i(p^*)$ (buyer $i$ is optimal) for all $i$.

**Market clearance**   By the complementary slackness condition regarding the optimal $(x^*, p^*)$, for item $j$ such that $\sum_i x^*_{ij} < 1$, it must holds that $p^*_j = 0$.

## A.2   Characterizations of Hoffman constant

We compare our definition of Hoffman constant and another common, explicit characterization. Recall that $H_{\mathcal{X}}(A)$ is the smallest $H$ such that, for any $b$, $\mathcal{S} = \{x : Ax = b\}$,

$$\|x - \Pi_{\mathcal{X} \cap \mathcal{S}}(x)\| \leq H \|Ax - b\|, \ \forall\, x \in \mathcal{X}.$$

For any matrix $M$, let $\mathcal{B}(M)$ be the set of *nonsingular* submatrices consisting of rows of $M$. Define

$$H(M) = \max_{B \in \mathcal{B}(M)} \frac{1}{\sigma_{\min}(B)} < \infty. \tag{9}$$

The following fact is known (see, e.g., [36, §11.8] and [4, §2.1]).

**Lemma 7** *Suppose the reference polyhedral set can be represented by inequality constraints $\mathcal{X} = \{x : Cx \leq d\}$. Then,*

$$H_{\mathcal{X}}(A) \leq H\left(\begin{bmatrix} A \\ C \end{bmatrix}\right).$$

Clearly, $H(M)$ is finite for any $M$. In fact, this is the most well-known characterization of Hoffman constant, and is tight in the following sense: let $\mathcal{S} = \{x : Ax = b\}$ for some arbitrary right hand side $b$, then it is the smallest constant $H$ such that

$$\|x - \Pi_{\mathcal{X} \cap \mathcal{S}}(x)\| \leq H \left\| \begin{bmatrix} Ax - b \\ (Cx - d)_+ \end{bmatrix} \right\|$$

for all $x$ (not necessarily $\in \mathcal{X}$). However, for all of our purposes, that is, analysis of PG, $x$ is always restricted to be $\in \mathcal{X}$. Therefore, we choose to define $H_{\mathcal{X}}(A)$ as such, consistent with [5] and [54]. Meanwhile, the following is clear.

**Lemma 8** *For any matrices $A \in \mathbb{R}^{m \times n}$, $m \leq n$ and $C \in \mathbb{R}^{\ell \times n}$, it holds that*

$$H\left(\begin{bmatrix} A \\ C \end{bmatrix}\right) \geq \max\left\{ \frac{1}{\sigma_{\min}(A)}, H(A) \right\}.$$

*Proof.* By definition (9), $H' := H\left(\begin{bmatrix} A \\ C^\top \end{bmatrix}\right) \geq H(A)$. If $\mathrm{rank}(A) = m$, then $H' \geq \frac{1}{\sigma_{\min}(A)}$ because $A \in \mathcal{B}\left(\begin{bmatrix} A \\ C \end{bmatrix}\right)$. If $r = \mathrm{rank}(A) < m$, let the (nonzero) singular values of $A$ be $\sigma_1 \geq \cdots \geq \sigma_r = \sigma_{\min}(A) > 0$. Consider any $B \in \mathcal{B}(A) \subseteq \mathcal{B}\left(\begin{bmatrix} A \\ C \end{bmatrix}\right)$ with rank $r$ (having exactly $r$ rows), let its nonzero singular values be $\sigma'_1 \geq \cdots \geq \sigma'_r = \sigma_{\min}(B) > 0$. Applying Cauhchy's Interlacing Theorem (see, e.g., [33, Theorem 1]) on $AA^\top$ and its principal submatrix $BB^\top$, we have

$$\sigma_1 \geq \sigma'_1 \geq \cdots \geq \sigma_r \geq \sigma'_r.$$

Therefore, $H' \geq \frac{1}{\sigma_{\min}(B)} \geq \frac{1}{\sigma_{\min}(A)}$. $\qquad\square$

## A.3   Proof of Theorem 2

We follow the development in [40, §4 & Appendix F] and further articulate the constants. There, the authors show that proximal gradient achieves linear convergence under the so-called *Proximal-PŁ* inequality. Consider the following general nonsmooth problem

$$F^* = \min_x F(x) = f(x) + g(x) \tag{10}$$

where $f$ is smooth convex with $L_f$-Lipschitz continuous gradient, $g$ is simple closed proper convex and $\mathrm{dom}\, g \subseteq \mathrm{dom}\, f$. One iteration of the proximal gradient method with stepsize $\gamma > 0$ is as follows:

$$x^{t+1} = \mathrm{Prox}_g\left(x^t - \gamma \nabla f(x^t)\right) = \arg\min_x \left[ \langle \gamma \nabla f(x), x - x^t \rangle + \frac{1}{2}\|x - x^t\|^2 + g(x) \right]. \tag{11}$$

For any $\alpha > 0$ and any $x \in \mathrm{dom}\, g$, define

$$\mathcal{D}(x, \alpha) = -2\alpha \min_{x'} \left[ \langle \nabla f(x), x' - x \rangle + \frac{\alpha}{2} \|x' - x\|^2 + g(x') - g(x) \right]. \tag{12}$$

Say that $F = f + g$ satisfies the proximal-PŁ inequality at $x$ w.r.t. $\Lambda \geq \lambda > 0$ if

$$\frac{1}{2}\mathcal{D}(x, \Lambda) \geq \lambda(F(x) - F^*), \tag{13}$$

Below is essentially [40, Theorem 5], which shows that the so-called Proximal-PŁ condition is sufficient for linear convergence. Note that, different from [40, Theorem 5], we only require (13) to hold for $x \in \mathcal{X}$ such that $F(x) \leq F(x^0)$ instead of all $x \in \mathcal{X}$. In addition, we note that in some cases (13) may hold with $\Lambda > L_f$, in which case the rate needs to be slightly adjusted. Since $\mathcal{D}(x, \cdot)$ is monotone [40, Lemma 1], (13) holds when $\Gamma$ is replaced by $\Gamma' \geq \Gamma$. The statement and proof are the same as [40, pp. 9] otherwise.

**Theorem 9** *Let $x^0 \in \mathrm{dom}\, g$. If $f$ and $g$ satisfies* (13) *for all $x \in \mathrm{dom}\, g$ such that $F(x) \leq F(x^0)$, then $x^t$ defined by* (11) *starting from $x^0$ with constant stepsize $\gamma = 1/L_f$ converges linearly with rate $1 - \frac{\lambda}{\bar{L}}$, where $\bar{L} = \max\{\Lambda, L_f\}$. In other words,*

$$F(x^t) - F^* \leq \left(1 - \frac{\lambda}{\bar{L}}\right)^t (F(x^0) - F^*), \ t = 1, 2, \ldots$$

*Proof.* By assumption, (13) holds for all $x \in \mathrm{dom}\, g$, $x \leq F(x^0)$. In particular, it holds for $x^t$, $t, 1, 2, \ldots$, since proximal gradient is a *descent* method, i.e., $F(x^0) \geq F(x^1) \geq \ldots$ (see, e.g., [3, Corollary 10.18]). Therefore, by $L_f$-Lipschitz continuity of $\nabla f$, proximal gradient update (11), definition of $D(x, \cdot)$, its monotonicity, and (13) for all $x^t$,

$$F(x^{t+1}) \leq F(x^t) + \langle \nabla f(x^t), x^{t+1} - x^t \rangle + \frac{L_f}{2}\|x^{t+1} - x^t\|^2 + g(x^{t+1}) - g(x^t)$$

$$\leq F(x^t) + \left[ \langle \nabla f(x^t), x^{t+1} - x^t \rangle + \frac{\bar{L}}{2}\|x^{t+1} - x^t\|^2 + g(x^{t+1}) - g(x^t) \right]$$

$$\leq F(x^t) - \frac{1}{2\bar{L}}\mathcal{D}(x^t, \bar{L})$$

$$\leq F(x^t) - \frac{\lambda}{\bar{L}}\left(F(x^t) - F^*\right)$$

$$\Rightarrow F(x^{t+1}) - F^* \leq \left(1 - \frac{\lambda}{\bar{L}}\right)\left(F(x^t) - F^*\right).$$

Repeatedly applying the above inequality completes the proof. $\qquad \square$

Then, we prove Theorem 2. Clearly, problem (2) is (10) with $g(x) = \delta_{\mathcal{X}}(x)$ and PG is a special case of proximal gradient. By Theorem 9, in order to prove Theorem 2, it suffices to establish the Proximal-PŁ condition (13) (for all $x \in \mathcal{X}$, $f(x) \leq f(x^0)$ for some initial iterate $x^0$). Let $\mathcal{X}^*$ be the set of optimal solutions to (2) and $f^*$ be the optimal objective value. Since $h$ is $\mu$-strongly convex and $f(x) = h(Ax)$, there exists $z^* \in \mathrm{dom}\, f$ such that $\mathcal{S} = \{x : Ax = z^*\}$ and $\mathcal{X}^* = \mathcal{X} \cap \mathcal{S}$. Therefore, for any $x \in \mathcal{X}$, $x_p := \Pi_{\mathcal{X}^*}(x)$, we have

$$f(x_p) = h(Ax_p) \geq h(Ax) + \langle \nabla h(Ax), A(x_p - x) \rangle + \frac{\mu}{2}\|A(x_p - x)\|^2.$$

Note that

$$\langle \nabla h(Ax), A(x_p - x) \rangle = \langle A^\top \nabla h(Ax), x_p - x \rangle = \langle \nabla f(x), x_p - x \rangle.$$

Hence, for any $x \in \mathcal{X}$, by strong convexity of $h$ and definition of $H = H_{\mathcal{X}^*}(A)$, we have

$$f(x_p) \geq f(x) + \langle \nabla f(x), x_p - x \rangle + \frac{\mu}{2}\|A(x - x_p)\|^2$$

$$= f(x) + \langle \nabla f(x), x_p - x \rangle + \frac{\mu}{2}\|Ax - z^*\|^2$$

$$\geq f(x) + \langle \nabla f(x), x_p - x \rangle + \frac{\mu}{2H^2}\|x - x_p\|^2,$$

Therefore,

$$\begin{aligned}
f^* &\geq f(x) + \langle \nabla f(x), x_p - x \rangle + \frac{\mu}{2H^2} \|x - x_p\|^2 \\
&\geq f(x) + \min_{y \in \mathcal{X}} \left\{ \langle \nabla f(x), y - x \rangle + \frac{\mu}{2H^2} \|y - x\|^2 \right\} \\
&\geq f(x) - \frac{H^2}{2\mu} \mathcal{D}\left(x, \frac{\mu}{H^2}\right) \\
&\Rightarrow \frac{1}{2} \mathcal{D}\left(x, \frac{\mu}{H^2}\right) \geq \frac{\mu}{H^2}(f(x) - f^*).
\end{aligned}$$

Thus, (13) holds for all $x \in \mathcal{X}$, $f(x) \leq f(x^0)$ with

$$\Lambda = \lambda = \frac{\mu}{H^2}.$$

Since $\nabla f(x) = A^\top \nabla h(Ax)$ and $h$ is $(\mu, L)$-s.c., its Lipschitz constant can be chosen as

$$L_f = L\|A\|^2.$$

By Theorem 9, PG with stepsize $\gamma = \frac{1}{L_f}$ converges linearly with rate

$$1 - \frac{\frac{\mu}{H^2}}{\max\left\{\frac{\mu}{H^2}, L\|A\|^2\right\}} = 1 - \frac{\mu}{\max\{\mu, LH^2\|A\|^2\}}.$$

Finally, convergence of the distance to optimality $\|x^t - \Pi_{\mathcal{X}^*}(x^t)\|$ is straightforward: for any $x \in \mathcal{X}$, by the strong convexity of $h$ and definition of $H$,

$$f(x) - f^* = h(Ax) - h(Ax_p) \geq \frac{\mu}{2}\|Ax - Ax_p\|^2 = \frac{\mu}{2}\|Ax - z^*\|^2 \geq \frac{\mu}{2H}\|x - x_p\|^2.$$

$\square$

**Remark** A special case is when $d \geq r$ (recall that $A \in \mathbb{R}^{d \times r}$) and $\text{rank}(A) = r$. In this case, $f(x) = h(Ax)$ itself is strongly convex with modulus $\mu\sigma_{\min}(A)^2$. In this case, classical analysis (e.g., [3, §10.6]) implies linear convergence with rate $1 - \frac{\mu\sigma_{\min}(A)^2}{L\|A\|^2}$. Meanwhile, in the above analysis, we have $\mathcal{X}^* = \{x^*\} = \mathcal{S} = \{x : Ax = z^*\} = \mathcal{X} \cap \mathcal{S}$ (since $x^*, z^*$ are unique and $\text{rank}(A) = r$). By Lemma 5, for any $x$, it holds that

$$\|x - \Pi_{\mathcal{X}^*}(x)\| \leq \frac{1}{\sigma_{\min}(A)^2}\|Ax - z^*\|.$$

Therefore, by the definition of Hoffman constant, $H_{\mathcal{X}^*}(A) \leq \frac{1}{\sigma_{\min}(A)^2}$ and the classical rate under strong convexity is recovered.

## A.4 Proof of Theorem 3

Let $\mathcal{X}^*$ be the set of optimal solutions to (2). First, recall the following lemma [65, Lemma 14], which ensures the first part of the theorem, that is, uniqueness of $Ax^*$ and $q^\top x^*$ for all $x^* \in \mathcal{X}^*$.

**Lemma 9** *There exist unique $z^* \in \mathbb{R}^r$ and $w^* \in \mathbb{R}$ such that for any $x^* \in \mathcal{X}^*$,*

$$Ax^* = z^*, \ \langle q^*, x \rangle = w^*.$$

The next lemma is essentially [4, Lemma 2.5]. Different from the statement of [4, Lemma 2.5], we keep $\|\nabla h(z^*)\|$ instead of bounding it by $\sup_{x \in \mathcal{X}} \|\nabla h(Ax)\|$. We also define $C = f(x^0) - f^*$ instead of $C = \sup_{x \in \mathcal{X}} f(x) - f^*$, since subsequent application of the lemma only involves PG iterates $x^t$, which have monotone decreasing objective values $f(x^0) \geq f(x^1) \geq \ldots$ The proof remains unchanged otherwise.

**Lemma 10** *Let $z^*$ be as in Lemma 9 and $x^0 \in \mathcal{X}$. For any $x \in \mathcal{X}$ such that $f(x) \leq f(x^0)$, it holds that*

$$\|x - \Pi_{\mathcal{X}^*}(x)\|^2 \leq \kappa \left( f(x) - f^* \right),$$

*where, same as in Theorem 3, $\kappa = H_{\mathcal{X}}(A)^2 \left( C + 2GD_A + \frac{2(G^2+1)}{\mu} \right)$, $C = f(x^0) - f^*$, $G = \|\nabla h(z^*)\|$, $D_A = \sup_{x,y \in \mathcal{X}} \|A(x-y)\|$.*

Finally, take $L_f = L\|A\|^2$ as a Lipschitz constant of $\nabla f$. By Lemma 10 and [40, §4.1], it holds that (2) satisfies the proximal-PŁ inequality (13) with

$$\Lambda = \lambda = \frac{1}{2\kappa}$$

for all $x \in \mathcal{X}$ such that $f(x) \leq f(x^0)$ (in particular, for all $x^t$, $t = 1, 2, \dots$). By Theorem 9, PG converges linearly with rate $1 - \frac{\lambda}{\max\{\Lambda, L_f\}} = 1 - \frac{1}{\max\{1, 2\kappa L\|A\|^2\}}$.

**Remark** Lemma 10 shows that QG holds. Similar convergence guarantees can also be derived from other QG-based analysis, e.g., [29, Corollary 3.7].

## A.5 Linear convergence of PG with linesearch

First, we consider the more general proximal gradient setup (10). Let $L_f$ be a Lipschitz constant of $\nabla f$ and the Proximal-PŁ inequality 13 holds with $\Lambda \geq \lambda \geq 0$ for all $x \in \mathrm{dom}\, g$ such that $F(x) \leq F(x^0)$. Let $\alpha \geq 1$, $\beta \in (0, 1)$, $\Gamma > 0$ (increment factor, decrement factor, upper bound on stepsize, respectively). The linesearch subroutine $\mathcal{LS}_{\alpha,\beta,\Gamma}$ is defined in Algorithm 1.

---

**Algorithm 1** $x_{t+1}, \gamma_t, k_t \leftarrow \mathcal{LS}_{\alpha,\beta,\Gamma}(x, \gamma, k_{\mathrm{prev}})$ with parameters $\alpha \geq 1$, $\beta \in (0, 1)$, $\Gamma > 0$.

If $k_{\mathrm{prev}} = 0$, set $\gamma^{(0)} = \min\{\alpha\gamma, \Gamma\}$. Otherwise, set $\gamma^{(0)} = \gamma$.
For $k = 0, 1, 2, \dots$
  1. Compute $x^{(k)} = \mathrm{Prox}_{\lambda^{(k)}g}(x - \gamma^{(k)}\nabla f(x))$.
  2. Break if

$$f(x^{(k)}) \leq f(x) + \langle \nabla f(x), x^{(k)} - x \rangle + \frac{1}{2\gamma^{(k)}}\|x^{(k)} - x\|^2. \qquad (14)$$

  3. Set $\gamma^{(k+1)} = \beta\gamma^{(k)}$ and continue to $k + 1$.
Return $x_{t+1} = x^{(k)}$, $\gamma_t = \gamma^{(k)}$, $k_t = k$.

---

In this way, proximal gradient with linesearch can be described formally as follows: starting from $x^0 \in \mathrm{dom}\, f$, $\gamma_{-1} = \Gamma$, $k_{-1} = 0$, perform the following iterations

$$(x^{t+1}, \gamma_t, k_t) \leftarrow \mathcal{LS}_{\alpha,\beta,\Gamma}(x^t, \gamma_{t-1}, k_{t-1}), \quad t = 1, 2, \dots$$

Note that (14) holds for any $\gamma^{(k)} \leq \frac{1}{L_f}$ (see, e.g., [3, Theorem 10.16]). Therefore, Algorithm 1 terminates when $\gamma^{(0)}\beta^k \leq \frac{1}{L_f}$. This means

$$\gamma_t \geq \tilde{\gamma} := \min\left\{\Gamma, \frac{\beta}{L_f}\right\}. \qquad (15)$$

for all $t$. Note that we explicitly include the case of $\Gamma \leq \frac{1}{L_f}$, although in practice $\Gamma$ is often set very large. Clearly,

$$\Gamma\beta^k \leq \tilde{\gamma} \Leftrightarrow k \geq \frac{\log \frac{\Gamma}{\tilde{\gamma}}}{\log \frac{1}{\beta}}.$$

Therefore, in Algorithm 1, the backtracking iteration index satisfies $k_t \leq \frac{\log \frac{\Gamma}{\tilde{\gamma}}}{\log \frac{1}{\beta}}$ for all $t$. Note that if the loop breaks at $k_t$, the number of $\mathrm{Prox}$ evaluations is exactly $k_t + 1$.

Let

$$\bar{L} = \max\left\{\frac{1}{\bar{\gamma}}, \Lambda\right\} = \max\left\{\frac{1}{\Gamma}, \frac{L_f}{\beta}, \Lambda\right\}. \tag{16}$$

Then, monotonicity of $D(x, \cdot)$ implies, for all $x \in \operatorname{dom} g$ such that $F(x) \leq F(x^0)$,

$$\frac{1}{2}\mathcal{D}(x, \bar{L}) \geq \frac{1}{2}\mathcal{D}(x, \Lambda) \geq \lambda\left(F(x) - F^*\right).$$

Following the proof of Theorem 9 (or that of [40, Theorem 5]), we have

$$F(x^{t+1}) \leq F(x^t) + \langle \nabla f(x^t), x^{t+1} - x^t \rangle + \frac{L_f}{2}\|x^{t+1} - x^t\|^2 + g(x^{t+1}) - g(x^t)$$

$$\leq F(x^t) + \langle \nabla f(x^t), x^{t+1} - x^t \rangle + \frac{\bar{L}}{2}\|x^{t+1} - x^t\|^2 + g(x^{t+1}) - g(x^t)$$

$$\leq F(x^t) - \frac{1}{2\bar{L}}\mathcal{D}\left(x^t, \bar{L}\right)$$

$$\leq F(x^t) - \frac{\lambda}{\bar{L}}(F(x^t) - F^*)$$

$$\Rightarrow F(x^{t+1}) - F^* \leq \left(1 - \frac{\lambda}{\bar{L}}\right)(F(x^t) - F^*).$$

Summarizing the above discussion, we have the following convergence guarantee for PG with linesearch.

**Theorem 10** *Let* $\alpha \geq 1$, $\beta \in (0, 1)$ *and* $\Gamma > 0$. *For problem* (10) *satisfying the Proximal-PŁinequality with* $\Lambda \geq \lambda > 0$ *for all* $x \in \operatorname{dom} g$ *such that* $F(x) \leq F(x^0)$, *proximal gradient* (11) *with linesearch subroutine* $\mathcal{LS}_{\alpha,\beta,\Gamma}$ *described in Algorithm 1 generates iterates* $x^t$ *such that*

$$F(x^{t+1}) - F^* \leq \left(1 - \frac{\lambda}{\bar{L}}\right)^t \left(F(x^0) - F^*\right), \quad t = 1, 2, \ldots, \tag{17}$$

*where* $\bar{L}$ *is defined in* (16). *Furthermore, each iteration requires at most* $1 + \frac{\log\frac{\Gamma}{\bar{\gamma}}}{\log\frac{1}{\beta}}$ *number of* $\operatorname{Prox}$ *evaluations.*

*Proof of Theorem 4.* In the above discussion, when $g(x) = \delta_{\mathcal{X}}(x)$, we can replace the Lipschitz constant $L_f$ by the restricted one $L_f^{\mathcal{X}}$ throughout, since Algorithm 1 ensures $x^t \in \mathcal{X}$ for all $t$. It remains to apply Theorem 10. For $q = 0$, $\Lambda = \lambda = \frac{\mu}{H^2}$ and $\bar{L} = \max\left\{\frac{1}{\Gamma}, \frac{L_f^{\mathcal{X}}}{\beta}, \frac{\mu}{H^2}\right\}$. Therefore, the rate is

$$1 - \frac{\lambda}{\bar{L}} = 1 - \frac{\mu}{\max\{\mu, H^2/\Gamma, H^2 L_f^{\mathcal{X}}/\beta\}}.$$

For $q \neq 0$, $\Lambda = \lambda = \frac{1}{2\kappa}$ and $\bar{L} = \max\left\{\frac{1}{\Gamma}, \frac{L_f^{\mathcal{X}}}{\beta}, \frac{1}{2\kappa}\right\}$. Therefore, the rate is

$$1 - \frac{1}{\max\{1, 2\kappa L_f^{\mathcal{X}}/\beta, 2\kappa/\Gamma\}}.$$

$\square$

## A.6 Other utility functions

Recall that, by Theorem (1), for any CCNH utilities $u_i$, optimal solutions to the EG convex program (1) correspond to equilibrium allocation and prices.

**CES utilities** are parametrized by a nondegenerate $v$ and exponent $\rho \in (-\infty, 1]\setminus\{0\}$:

$$u_i(x_i) = \left(\sum_{j=1}^m v_{ij} x_{ij}^\rho\right)^{1/\rho}.$$

Clearly, $\rho = 1$ gives linear utilities. For $\rho < 1$, it has been shown that Proportional Response dynamics achieves linear convergence in prices and utilities [68, Theorem 4] under their notion of $\epsilon$-approximate market equilibrium [68, pp. 2693].

**Cobb-Douglas utilities** represent substitutive items and take the following form, for parameters $\lambda = (\lambda_i), \lambda_i \in \Delta_m$:

$$u_i(x_i) = \Pi_j x_{ij}^{\lambda_{ij}}.$$

In this case, EG (1) decomposes item-wise into simple problems with explicit solutions. Specifically, for each item $j$, the minimization problem is

$$\min_{x_{:,j} \in \Delta_n} -\sum_i B_i \lambda_{ij} \log x_{ij}.$$

Let $p_j$ be the Lagrangian multiplier associated with constraint $\sum_i x_{ij} = 1$. The Lagrangian is

$$\mathcal{L}(x_{:,j}, p_j) = -\sum_i B_i \lambda_{ij} \log x_{ij} + p_j \left( \sum_i x_{ij} - 1 \right).$$

By first-order stationarity condition, for any $p_j \in \mathbb{R}$, $\mathcal{L}(x_{:,j}, p_j)$ is minimized when

$$x_{ij} = \frac{B_i \lambda_{ij}}{p_j}. \tag{18}$$

Substituting it into $\mathcal{L}$ and discarding the constants w.r.t. $p_j$, we have

$$g(p_j) = \left( \sum_i B_i \lambda_{ij} \right) \log p_j - p_j,$$

which is maximized at equilibrium prices

$$p_j^* = \sum_i B_i \lambda_{ij}.$$

Therefore, by 18, the equilibrium $x^*$ under Cobb-Douglas utilities is given by

$$x_{ij}^* = \frac{B_i \lambda_{ij}}{\sum_i B_i \lambda_{ij}}, \ \forall i, j.$$

## B   Linear utilities

### B.1   Shmyrev's convex program

Under linear utilities, it turns out that we can also compute market equilibrium via the following convex program due to Shmyrev [59, 9]. In this convex program, the variables are the *bids* $b_{ij}$, $i \in [n]$, $j \in [m]$ and prices $p_j$, $j \in [m]$.

$$\max \sum_{i,j} b_{ij} \log v_{ij} - \sum_j p_j \log p_j \ \text{ s.t. } \sum_i b_{ij} = p_j, \ j \in [m], \ \sum_j b_{ij} = B_i, \ i \in [n], \ b \geq 0. \tag{19}$$

Given an optimal solution $b^*$, equilibrium prices and allocations are then given by $p_j^* = \sum_i b_{ij}^*$ and $x_{ij}^* = \frac{b_{ij}^*}{p_j^*}$, respectively.

### B.2   Proof of Lemma 1

Any $x \in \mathcal{X}$ satisfies $x \leq 1$. Therefore, $\langle v_i, x_i \rangle \leq \|v_i\|_1 \|x_i^*\|_\infty \leq \|v_i\|_1 = \bar{u}_i$. For the lower bound, recall that at an equilibrium allocation $x^*$ ensures that every buyer gets at least the utility of the proportional share, that is,

$$\langle v_i, x_i^* \rangle \geq \left\langle v_i, \frac{B_i}{\|B\|_1} \mathbf{1} \right\rangle = \frac{B_i \|v_i\|_1}{\|B\|_1} = \underline{u}_i.$$

## B.3 Uniqueness of equilibrium quantities and convergence of $u^t, p^t$

Convergence of $u^t$ to $u^*$ can be easily seen as follows. Let $x^t$ be the PG iterates and $\tilde{h}$, $A$, $f = \tilde{h}(Ax)$, $\mu$ be defined as in §3 and $f^* = \min_{x \in \mathcal{X}} f(x)$. Since $\tilde{h}$ is $\mu$-strongly convex, we have

$$\frac{\mu}{2}\|u^t - u^*\|^2 \leq \tilde{h}(u^t) - \tilde{h}(u^*) \leq \tilde{h}(Ax^t) - f^*,$$

which converges linearly. Next, we show uniqueness of $p^*$ via simple arguments and construct a sequence of linearly convergent prices $p^t$.

**Lemma 11** *Assume that $v$ is nondegenerate. Then, the equilibrium prices $p^*$ under linear utilities are unique.*

*Proof.* By Theorem 1 and [22, Lemma 3], $p^*$ is an optimal solution (together with some $\beta^*$) to the following problem (dual of (1) with linear utilities): $\qquad\qquad\square$

$$\min_{p, \beta} \sum_j p_j - \sum_i B_i \log \beta_i \ \text{ s.t. } p \geq 0, \ \beta \geq 0, \ p_j \geq v_{ij}\beta_i, \ \forall\, i, j. \qquad (20)$$

Here, strong duality holds since there clearly exist primal and dual strictly feasible solutions with finite objective values given nondegenerate $v$ (c.f. Theorem 1 and Appendix A.1). We can eliminate $p$ by letting $p_j = \max_i v_{ij}\beta_i$ for all $j$ and rewrite (20) as

$$\min_{\beta} \sum_j \max_i v_{ij}\beta_j - \sum_i B_i \log \beta_i \ \text{ s.t. } \beta \geq 0.$$

In the above, since the objective is strongly convex and the feasible region is $\beta \geq 0$, the optimal solution $\beta^*$ is clearly unique. Furthermore, it must hold that $\beta^* > 0$ (since the optimal objective value is finite and strong duality holds). For $p^*$ optimal to (20), it must hold that $p_j^* = \max_i v_{ij}\beta_i^*$. In fact, $p_j^* \geq \max_i v_{ij}\beta_i^*$ by feasibility and, for any strict inequality, decreasing the corresponding $p_j^*$ strictly decreases the objective. $\qquad\qquad\square$

The following lemma provides simple upper and lower bounds on feasible and equilibrium prices, respectively. The lower bounds are slightly strengthened over the existing one [9, Lemma 17].

**Lemma 12** *Let $p^*$ be equilibrium prices under linear utilities with nondegenerate valuations $v$. Then, $\underline{p}_j \leq p_j^* \leq \bar{p}_j$ for all $j$, where $\underline{p}_j = \max_i \frac{v_{ij}B_i}{\|v_i\|_1}$ and $\bar{p}_j = \|B\|_1$.*

*Proof.* It is essentially the same as the proof of Lemma 2, except that, at optimality, $u_i \leq \|v_i\|_1 + B_i$ can be strengthened to $u_i \leq \|v_i\|_1$ (utility of each buyer is at most that of having a unit of every item). $\square$

**A linearly convergent sequence of $p^t$.** Here, all norms are vector norms. Note that each step of PG is of the form $x^{t+1} = \Pi_{\mathcal{X}}(\bar{x}^t)$, where $\bar{x}^t = x^t - \gamma \nabla f(x^t)$. Since $\nabla f$ is $L_f$-Lipschitz, the mapping

$$\phi_1 : x \mapsto x - \gamma \nabla f(x^t)$$

is Lipschitz continuous (w.r.t. $\|\cdot\|_2$) with constant $1 + \gamma L_f = 2$ (where $\gamma = \frac{1}{L\|A\|^2}$ is the fixed stepsize). Meanwhile, we have the following.

**Lemma 13** *Let $y \in \mathbb{R}^n$ and $y^* = \Pi_{\Delta^n}(y)$. There exists a unique multiplier $\lambda \in \mathbb{R}$, which can be computed in $O(n \log n)$ time, such that*

$$\sum_{i=1}^n (y_i - \lambda)_+ = 1. \qquad (21)$$

*Moreover, the mapping $\phi_2 : y \mapsto \lambda$ is piecewise linear and 1-Lipschitz continuous w.r.t. $\|\cdot\|_1$.*

*Proof.* By the KKT conditions for simplex projection (see, e.g., [66, §3]), it holds that there exists unique $\lambda$ such that

$$y^* = (y - \lambda\mathbf{1})_+.$$

Suppose there exists $\lambda_1 < \lambda_2$ that satisfy (21). Then, since the left-hand side of (21), denoted as $w(\lambda)$, is monotone decreasing in $\lambda$, it must hold that $w(\lambda) = 1$ for all $\lambda \in [\lambda_1, \lambda_2]$. In other words, $w(\cdot)$ is *constant* on $[w_1, w_2]$. This further implies $w(\lambda) = 0$ for all $w \in [w_1, w_2]$, a contradiction. Therefore, $\lambda = \phi_2(y)$ is uniquely defined. Let $I^+(y)$, $I^0(y)$, $I^-(y)$ denote the set of indices $i \in [n]$ such that $y_i > \lambda$, $y_i = \lambda$, $y_i < \lambda$, respectively (where $\lambda = \phi_2(y)$). We have

$$\lambda = \frac{\sum_{i \in I^+(y)} y_i - 1}{|I^+(y)|} = \frac{\sum_{i \in I^+(y) \cup I^0(y)} y_i - 1}{|I^+(y)| + |I^0(y)|},$$

which is piecewise linear in $y$ since there are only finitely many index possible sets of indices and $I^+(y)$ is always nonempty (otherwise $\sum_i (y_i - \lambda)_+ = 0$). To see Lipschitz continuity, let $y'$ be such that $\|y' - y\|_1 \le \epsilon$, where $0 < \epsilon < \min\{|y_i - y_j| : i, j \in [n], y_i \ne y_j\}$. It must hold that $I^+(y) \subseteq I^+(y')$. In other words, $\lambda' = \phi_2(y')$ does not deactivate any $i \in I^+(y)$, only bringing new $i \in I^0(y)$. Hence, it holds that $|\lambda' - \lambda| \le \frac{\|y - y'\|_1}{|I^+(y)|} \le \|y - y'\|_1$. In other words, $\phi_2$ is 1-Lipschitz continuous w.r.t. $\|\cdot\|_1$.

Finally, [66, Algorithm 1]) computes $\lambda$ and $y^*$ in $O(n \log n)$ time. $\qquad \square$

Abusing the notation, let $\phi_2$ also denote the mapping from $x \in \mathbb{R}^{n \times m}$ to $\lambda \in \mathbb{R}^m$, that is, $\lambda_j = \varphi_2(x_{1j}, \ldots, x_{nj})$. Let

$$\phi(x) = \phi_2(\phi_1(x))/\gamma$$

and $p^t = \phi(x^t)$. Here, $\phi_1$ is 2-Lipschitz continuous and $\phi_2$ is 1-Lipschitz continuous w.r.t. $\|\cdot\|_1$. For any optimal solution $x^* \in \mathcal{X}^*$, by $x^* = \Pi_{\mathcal{X}}(x^*)$ and KKT conditions for (1) and (20), it can be seen that

$$p^* = \phi(x^*).$$

Using the Lipschitz continuity properties of $\phi_1, \phi_2$ and Theorem 2, we have

$$\|p^t - p^*\|_1 = \|\phi(x^t) - \phi(\Pi_{\mathcal{X}^*}(x^t))\|_1 \le \frac{1}{\gamma}\|\phi_1(x^t) - \phi_1(\Pi_{\mathcal{X}^*}(x^t))\|_1$$

$$\le \frac{n}{\gamma}\|\phi_1(x^t) - \phi_1(\Pi_{\mathcal{X}^*}(x^t))\| \le \frac{2n}{\gamma} \cdot \|x^t - \Pi_{\mathcal{X}^*}(x^t)\|$$

$$\le \frac{2n}{\gamma} \cdot \sqrt{\frac{2H_{\mathcal{X}}(A)}{\mu} \left(f(x^t) - f^*\right)}$$

$$\le \frac{2n}{\gamma}\sqrt{\frac{2H_{\mathcal{X}}(A)}{\mu}} \cdot \left(1 - \frac{\mu}{2HL\|A\|^2}\right)^{t/2} \cdot \sqrt{f(x^0) - f^*}.$$

Therefore, we can take $C = \frac{2n}{\gamma}\sqrt{\frac{2H_{\mathcal{X}}(A)}{\mu}} \cdot \sqrt{f(x^0) - f^*}$ and $\rho = \sqrt{1 - \frac{\mu}{2HL\|A\|^2}} \in (0, 1)$.

Since $p^* \ge \underline{p} > 0$, we can bound the maximum relative price error $\eta^t = \max_j \frac{|p_j^t - p_j^*|}{p_j^*}$ as follows, where $\underline{p}_{\min} = \min_j \underline{p}_j$.

$$\eta^t \le \frac{\|p^t - p^*\|_1}{\underline{p}_{\min}} \le \frac{C}{\underline{p}_{\min}} \cdot \rho^t.$$

In other words, $\eta^t$ converges (R-)linearly to zero.

## C  QL utilities

### C.1  Derivation of the QL-Shmyrev convex program (4)

In [22, Lemma 5], the convex program for the equilibrium prices is as follows:

$$\min \sum_j p_j - \sum_i B_i \log \beta_i \ \text{ s.t. } v_{ij}\beta_i \le p_j, \ \forall i, j, \ 0 \le \beta \le 1. \tag{22}$$

Note that it is simply the dual of EG under linear utilities (20) with additional constraints $\beta \le 1$. Assuming $v$ is nondegenerate, by a change of variable and Lagrange duality, we can derive the dual

of (22). First, at optimality, it must holds that $\beta_i > 0$ for all $i$. Therefore, by nondegeneracy of $v$, $p_j > 0$ for all $j$ at optimality. Let $p_j = e^{q_j}$ and $\beta_i = e^{-\gamma_i}$. The above problem is equivalent to

$$
\begin{aligned}
\min \ & \sum_j e^{q_j} + \sum_i B_i \gamma_i \\
\text{s.t.} \ & q_j + \gamma_i \geq \log v_{ij}, \ \forall\, i, j, \\
& \gamma \geq 0.
\end{aligned}
\tag{23}
$$

Let $b_{ij} \geq 0$ be the dual variable associated with constraint $q_j + \gamma_i \geq \log v_{ij}$. The Lagrangian is

$$
\begin{aligned}
L(q, \gamma, b) &:= \sum_j e^{q_j} + \sum_i B_i \gamma_i - \sum_{i,j} b_{ij} \left( q_j + \gamma_i - \log v_{ij} \right) \\
&= \sum_j \left( e^{q_j} - \left( \sum_i b_{ij} \right) q_j \right) + \sum_i (B_i - \sum_j b_{ij}) \gamma_i + \sum_{i,j} (\log v_{ij}) b_{ij}.
\end{aligned}
$$

Clearly, when $\sum_j b_{ij} \leq B_i$ for all $i$, $\gamma \geq 0$, $L(q, \gamma, b)$ is minimized at $q_j = \log \sum_i b_{ij}$ and $\gamma = 0$. When $\sum_j b_{ij} > B_i$ for some $i$, $L \to -\infty$ as $\gamma_i \to \infty$. Therefore, when $\sum_j b_{ij} \leq B_i$ for all $i$, we have

$$
g(b) = \sum_j \left[ \sum_{i,j} b_{ij} - \left( \sum_i b_{ij} \right) \log \sum_i b_{ij} \right] + \sum_{i,j} (\log v_{ij}) b_{ij}.
$$

Therefore, the dual is

$$
\max g(b) \ \text{s.t.} \ b \geq 0, \ \sum_j b_{ij} \leq B_i, \ \forall\, i.
$$

Adding slack variables $\delta = (\delta_1 \ldots, \delta_n)$ and writing it in minimization form yield (4).

**Remark.** As mentioned in §4, when some $v_{ij} = 0$ (but $v$ is still nondegenerate), by the above derivation, the first summation in (4) should be replaced by $\sum_{(i,j) \in \mathcal{E}}$, where $\mathcal{E} = \{(i,j) : v_{ij} > 0\}$. The dual remains the same otherwise.

## C.2 Proof of Lemma 2

Similar to the proof of Lemma 11, this can be seen via the uniqueness of the optimal solution $(p^*, \beta^*)$ of (22), that is, from uniqueness of $\beta^*$ to that of $p_j^* = \max_i v_{ij} \beta_i^*$.

Let $(b^*, \delta^*)$ be an optimal solution to (4). Note that strong duality holds for (23) and (4), since there exit simple strictly feasible solutions. By the derivation in Appendix C.1, it holds that $q_j^* = \log \sum_i b_{ij}^*$ gives an optimal solution to (23) (the first-order optimality condition). Therefore,

$$
p_j^* = e^{q_j^*} = \sum_i b_{ij}^*.
$$

Next we establish the upper and lower bounds on $p^*$. By the derivation in Appendix C.1 and Lagrange duality, for any optimal solution $b^*$ to (4), it holds that $p_j^* := \sum_i b_{ij}^*$ and $\beta_j^* = \min_{j \in J_i} \frac{p_j^*}{v_{ij}}$ give the (unique) optimal solution to (22). Clearly, $\beta^* \leq 1$ and therefore

$$
p_j^* = \max_i v_{ij} \beta_i^* \leq \max_i v_{ij} = \bar{p}_j.
$$

By [22, Lemma 5], the dual of (22) is (c.f. the original EG primal 1)

$$
\begin{aligned}
\max_{u,\, x,\, s} \ & \sum_i B_i \log u_i - s_i \\
\text{s.t.} \ & u_i \leq v_i^\top x_i + s_i, \ \forall\, i, \\
& \sum_i x_{ij} \leq 1, \ \forall\, j, \\
& x, s \geq 0.
\end{aligned}
\tag{24}
$$

Clearly, strong duality holds for (22) and (24). Furthermore, notice the following.

- $\beta_i^* = \frac{B_i}{u_i^*}$ at optimality, where $u_i^*$ is the amount of utility of buyer $i$. This is by the stationarity condition in the KKT optimality conditions.
- $u_i^* \le \|v_i\|_1 + B_i$, where the right hand side is the amount of utility of all items and the entire budget. This can also be seen as follows. When $s_i > B_i$, decreasing $s_i$ strictly increases the objective of (24). Therefore, the optimal $s^*$ must satisfy $s_i^* \le B_i$. It then follows from the constraint $u_i \le v_i^\top x_i + s_i$.

Therefore,
$$p_j^* \ge \max_i v_{ij}\beta_i^* \ge \max_i \frac{v_{ij}B_i}{\|v_i\|_1 + B_i} = \underline{p}_j.$$

### C.3 Proof of Theorem 7

Similar to [9, Lemma 7], we first establish the following "generalized Lipschitz condition" for $\varphi$, which is key to the claimed last-iterate convergence.

**Lemma 14** *For all $(b, \delta), (b', \delta') \in \mathcal{B}$, it holds that*
$$\varphi(b') \le \varphi(b) + \langle \nabla\varphi(b), b' - b \rangle + D(b', \delta' \| b, \delta). \tag{25}$$

*Proof.* Recall that $p_j(b) = \sum_i b_{ij}$, $\frac{\partial}{\partial b_{ij}}\varphi(b) = \log \frac{p_j(b)}{v_{ij}}$. For $(a, \delta^a), (b, \delta^b) \in \mathcal{B}$, we have
$$
\begin{aligned}
&\varphi(b) - \varphi(a) - \langle \nabla\varphi(a), b - a \rangle \\
&= -\sum_{i,j}(1 + \log v_{ij})(b_{ij} - a_{ij}) + \sum_j p_j(b)\log p_j(b) - \sum_j p_j(a)\log p_j(a) \\
&\quad - \sum_{i,j}(b_{ij} - a_{ij})\log\frac{p_j(a)}{v_{ij}} \\
&= -\sum_{i,j}(b_{ij} - a_{ij}) + \sum_j p_j(b)\log\frac{p_j(b)}{p_j(a)} \\
&= \sum_i(\delta_i^b - \delta_i^a) + \sum_j p_j(b)\log\frac{p_j(b)}{p_j(a)}. \tag{26}
\end{aligned}
$$
Note that convexity and smoothness of $x \mapsto x\log\frac{x}{y}$ $(y > 0)$ implies
$$\delta_i^b - \delta_i^a \le \delta_i^b \log\frac{\delta_i^b}{\delta_i^a}. \tag{27}$$
As in the proof of [9, Lemma 7], by convexity of $q(x, y) = x\log\frac{x}{y}$, it holds that
$$\sum_j p_j(b)\log\frac{p_j(b)}{p_j(a)} \le \sum_{i,j} b_{ij}\log\frac{b_{ij}}{a_{ij}}. \tag{28}$$
By (27) and (28), the right hand side of (26) can be bounded by $D(b, \delta^b \| a, \delta^a)$. Therefore, (25) holds. $\square$

Next, we prove the inequality on the right. Clearly, $(b^0, \delta^0) \in \mathcal{B}$. By [9, Theorem 3] (with objective $f = \varphi$, constraint set $C = \mathcal{B}$ and stepsize $\gamma$), we have
$$\varphi(b^t) - \varphi(b^*) \le \frac{D(b^*, \delta^* \| b^0, \delta^0)}{t}.$$
Similar to the proof of [9, Lemma 13], we can bound the Bregman divergence on the right hand side as follows, where $b_{ij} = \delta_i = \frac{B_i}{m+1}$.
$$
\begin{aligned}
D(b^*, \delta^* \| b^0, \delta^0) &= \sum_{i,j} b_{ij}^* \log\frac{b_{ij}^*}{B_i} + \sum_i \delta_i^* \log\frac{\delta_i^*}{B_i} + \sum_{i,j} b_{ij}^* \log(m+1) + \sum_i \delta_i^* \log(m+1) \\
&\le \sum_{i,j} b_{ij}^* \log(m+1) + \sum_i \delta_i^* \log(m+1) \\
&\le \|B\|_1 \log(m+1),
\end{aligned}
$$

where the first inequality is because $\frac{b_{ij}^*}{B_i} \le 1$. Combining the above yields the desired inequality.[1]

Finally, we show the inequality on the left. By optimality of $(b^*, \delta^*)$, we have

$$\langle \nabla \varphi(b^*), b - b^* \rangle \ge 0, \ \ \forall (b, \delta) \in \mathcal{B}.$$

Recall that $p_j(b) = \sum_i b_{ij}$. By (26), we have

$$D(p^t \| p^*) = -\sum_{i,j}(b_{ij}^t - b_{ij}^*) + \sum_j p_j(b^t) \log \frac{p_j(b^t)}{p_j(b^*)} \le \varphi(b^t) - \varphi^*.$$

$\square$

## C.4 Details from MD (5) to PR (6)

Note that (5) is buyer-wise separable: for each $i$, we have (where $\frac{\partial}{\partial b_{ij}} \varphi_b(b) = \log \frac{p_j(b)}{v_{ij}}$ and $\mathcal{B}_i = B_i \cdot \Delta_{m+1}$)

$$(b_i^{t+1}, \delta_i^{t+1}) = \arg\min_{(b_i, \delta_i) \in \mathcal{B}_i} \sum_j \left( \log \frac{p_j(b^t)}{v_{ij}} - \log b_{ij}^t \right) b_{ij} - (\log \delta_i^t)\delta_i + \sum_j b_{ij} \log b_{ij} + \delta_i \log \delta_i$$

$$= \arg\min_{(b_i, \delta_i) \in \mathcal{B}_i} -\sum_j (\log b_{ij}^t) b_{ij} - (\log \delta_i^t)\delta_i + \sum_j b_{ij} \log b_{ij} + \delta_i \log \delta_i. \tag{29}$$

By Lemma 6, for all $i, j$,

$$b_{ij}^{t+1} = B_i \cdot \frac{\frac{v_{ij} b_{ij}^t}{p_j(b^t)}}{\sum_\ell \frac{v_{i\ell} b_{i\ell}^t}{p_\ell(b^t)} + \delta_i^t}, \quad \delta_j^{t+1} = B_i \cdot \frac{\delta_i^t}{\sum_\ell \frac{v_{i\ell} b_{i\ell}^t}{p_\ell(b^t)} + \delta_i^t}. \tag{30}$$

Let $p_j^t = p_j(b^t)$. Then, (30) can be written in terms of the allocations $x_{ij}^t = b_{ij}^t / p_j^t$ (which sum up to 1 over buyers $i$ for any item $j$) and leftover $\delta_i^t$, thus giving (6).

## C.5 Convergence of prices

Let $\eta^t = \max_j \frac{|p_j^t - p_j^*|}{p_j^*}$ be the relative price error, which can clearly be bounded by $\frac{\|p^t - p^*\|_1}{p_{\min}}$, where $p_{\min} = \min_j p_j > 0$ is given in Lemma 2. By Theorem 7 and strong convexity of KL divergence (w.r.t. $\| \cdot \|_1$), for $b^t$ and $p^t = p(b^t)$ generated by either PG or PR,

$$\frac{1}{2}\|p^t - p^*\|_1^2 \le D(p^t \| p^*) \le \varphi(b^t) - \varphi^*. \tag{31}$$

Therefore, for PG, the quantities $\eta^t$, $\|p^t - p^*\|$ and $D(p^t \| p^*)$ all converge linearly to 0. For PR, they converge at $O(1/T)$.

We can further bound $\varphi(b^t) - \varphi^*$ by the duality gap. Specifically, given $b^t$, $p^t = p(b^t)$, let

$$b_i^t = \min \left\{ \min_j \frac{p_j^t}{v_{ij}}, 1 \right\}.$$

Then, $(p^t, \beta^t)$ is feasible to (22). By weak duality,

$$\varphi(b^t) - \varphi^* \le \varphi(b^t) + g(p^t, \beta^t), \tag{32}$$

where $g(p, \beta)$ is the (minimization) objective of (22). Combining the above, we have

$$\eta^t \le \frac{\sqrt{2\left(\varphi(b^t) + g(p^t, \beta^t)\right)}}{p_{\min}}.$$

Note that the above holds for $b^t$ from either PG or PR. Although neat in theory, numerical experiments suggest that the above bound can be loose and is not suitable as a termination criteria.

# D   Leontief utilities

## D.1   Derivation of (7)

The primal EG (1) under Leontief utilities $u_i(x_i) = \min_{j \in J_i} \frac{x_{ij}}{a_{ij}}$ can be written in both $x$ and $u$:

$$\min_{u,\, x} \; - \sum_i B_i \log u_i$$

$$\text{s.t. } u_i \leq \frac{x_{ij}}{a_{ij}}, \; \forall j \in J_i, \; \forall i \in [n],$$

$$\sum_i x_{ij} \leq 1, \; \forall j \in [m],$$

$$x \geq 0, \; u \geq 0.$$

Clearly, it can also be written in terms of $u_i$ only as follows:

$$\min \; - \sum_i B_i \log u_i \; \text{ s.t. } \sum_{i \in I_j} a_{ij} u_i \leq 1, \; \forall j, \; u \geq 0. \tag{33}$$

Let $p_j \geq 0$ be the dual variable associated with constraint $\sum_{i \in I_j} a_{ij} u_i \leq 1$. The Lagrangian is

$$\mathcal{L}(u, p) = - \sum_i B_i \log u_i + \sum_j p_j \left( \sum_{i \in I_j} a_{ij} u_i - 1 \right)$$

$$= - \sum_j p_j + \sum_i \left[ -B_i \log u_i + \langle a_i, p \rangle u_i \right].$$

Note that minimizing $\mathcal{L}$ w.r.t. $u$ can be performed separably for each $u_i$. For any $i$ such that $\sum_{j \in J_i} p_j > 0$, by first-order stationarity condition, the term $-B_i \log u_i + \langle a_i, p \rangle u_i$ is minimized at $u_i^*(p) = \frac{B_i}{\langle a_i, p \rangle}$ with minimum value $B_i(1 - \log B_i) + B_i \log \langle a_i, p \rangle$. If $\sum_{j \in J_i} p_j = 0$, the term approaches $-\infty$ as $u_i \to \infty$. Therefore, the dual objective is

$$g(p) = \begin{cases} - \sum_j p_j + \sum_i B_i \log \langle a_i, p \rangle + \sum_i B_i (1 - \log B_i) & \text{if } p \geq 0 \text{ and } \sum_{j \in J_i} a_{ij} p_j > 0 \\ -\infty & \text{o.w.} \end{cases}$$

Hence the (Lagrangian) dual problem is $\max_p g(p)$. Its minimization form, up to the constant $-\sum_i B_i(1 - \log B_i)$, is

$$\min \left[ \sum_j p_j - \sum_i B_i \log \langle a_i, p \rangle \right] \quad \text{s.t. } p \geq 0. \tag{34}$$

By Theorem 1, we have the following.

- An optimal solution to (34) gives equilibrium prices.
- A market equilibrium $(x^*, p^*)$ satisfies $\langle p^*, x_i \rangle = B_i$ for all $i$ and $\sum_i x_{ij}^* = 1$ for all $j$. Therefore, we have $\sum_j p_j^* = \|B\|_1$.

Therefore, we can add the constraint $\sum_j p_j = \|B\|_1$ to (34) without affecting any optimal (equilibrium) solution. This leads to (7).

## D.2   Proof of Lemma 3

let $p$ be any feasible solution to (7). Since $\sum_j p_j = \|B\|_1$, we have $\langle a_i, p \rangle \leq \|a_i\|_\infty \|p\|_1 = \|a_i\|_\infty \|B\|_1$ for all $i$. By Appendix D.1, at equilibrium, $p^*$ and primal variables $u_i^*$ satisfy $u_i^* = \frac{B_i}{\langle a_i, p^* \rangle}$ (the stationarity condition) and

$$u_i^* \leq \text{utility of getting one unit of every item} = \min_{j \in J_i} \frac{1}{a_{ij}} = \frac{1}{\|a_i\|_\infty}$$

for all $i$. Therefore $\langle a_i, p^* \rangle = \frac{B_i}{u_i^*} \leq \|a_i\|_\infty \|B\|_1$.

## D.3 Linear convergence of utilities

Note that the equilibrium utilities $u^*$ are clearly unique by (33). By the KKT stationary condition,

$$u_i^* = \frac{B_i}{\langle a_i, p^* \rangle}, \quad \forall i$$

for equilibrium prices $p^*$. Therefore, an intuitive construction of $u^t$ is as follows. Let $p^t$ be the current iterate, $r_i^t = \langle a_i, p \rangle$. First compute $\tilde{u}_i^t = \frac{B_i}{r_i^t}$. Then, to satisfy the primal constraints $\sum_i u_i a_{ij} \leq 1$, take

$$u^t = \frac{\tilde{u}^t}{\max_j \sum_i u_i a_{ij}} = \frac{\tilde{u}^t}{\|a^\top \tilde{u}\|_\infty}.$$

Let $r^* = \langle a_i, p^* \rangle = \frac{B_i}{u_i^*}$ and $f^* = \arg\min_{p \in \mathcal{P}} \tilde{h}(ap) = \tilde{h}(r^*) = h(r^*)$. Strong convexity of $\tilde{h}$ implies $\frac{\mu}{2} \|r^t - r^*\|^2 \leq h(r^t) - f^*$. Furthermore, the mapping $r^t \mapsto \tilde{u}^t \mapsto u^t$ is Lipschitz continuous on $r^t \in [\underline{r}, \bar{r}]$. Therefore, $\|u^t - u^*\|$ converges to 0 linearly as well.

# E  Additional details on numerical experiments

For linear utilities, we generate market data $v = (v_{ij})$ where $v_{ij}$ are i.i.d. from standard Gaussian, uniform, exponential, or lognormal distribution. For each of the sizes $n = 50, 100, 150, 200$ (on the horizontal axis) and $m = 2n$, we generate 30 instances with unit budgets $B_i = 1$ and random budgets $B_i = 0.5 + \tilde{B}_i$ (where $\tilde{B}_i$ follows the same distribution as $v_{ij}$). See §6 for plots under random budgets and below for those under uniform budgets.

The termination conditions (on the vertical axis) are

$$\epsilon(p^t, p^*) \leq \eta, \ \eta = 10^{-2}, 10^{-3},$$

where $p^*$ is the optimal Lagrange multipliers of (1) computed by CVXPY+Mosek. Then, for $n = 100, 200, 300, 400$ and $n = 2m$, we repeat the above with termination conditions

$$\mathrm{dgap}_t / n \leq \eta, \ \eta = 10^{-3}, 10^{-4}, 10^{-5}, 5 \times 10^{-6}.$$

For QL utilities, we repeat the above (same random $v$, same sizes and termination conditions) using budgets $B_i = 5(1 + \tilde{B}_i)$. This is to make buyers have nonzero bids and leftovers (i.e., $0 < \delta_i^* < B_i$) at equilibrium in most scenarios. In this case, $p^* = p(b^*)$, where $b^*$ is the optimal solution to (4) computed by CVXPY+Mosek. For QL, FW does not perform well in initial trials and is excluded in subsequent experiments.

For the linesearch subroutine $\mathcal{LS}_{\alpha, \beta, \Gamma}$ in PG (see Appendix A.5), we use parameters $\alpha = 1.02$, $\beta = 0.8$ and $\Gamma = 100L\|A\|^2$ throughout.

For Leontief utilities, in addition to $\mathrm{dgap}_t / n \leq \eta$, we also use the termination condition $\epsilon(u^t, u^*) = \max_j \frac{|u_j^t - u_j^*|}{u_j^*} \leq \eta$, where $u^*$ is the optimal solution to EG under Leontief utilities (33) computed by CVXPY+Mosek.

**Computing the duality gap.** For linear utilities, the objective of the original Shmyrev's convex program (19) is

$$\varphi(b) = -\sum_{i,j} (\log v_{ij}) b_{ij} + \sum_j p_j(b) \log p_j(b)$$

where $p_j(b) = \sum_i b_{ij}$. Recall the objective of the (EG) dual (20), equivalent to the dual of Shmyrev's (19),

$$g(p, \beta) = \sum_j p_j - \sum_i B_i \log \beta_i.$$

Given iterate $b^t$, let $p_j^t = p_j(b^t)$ and $\beta_i^t = \min_j \frac{p_j}{v_{ij}}$, which is finite since $v$ is nondegenerate and $p^t > 0$. The duality gap is computed via

$$\mathrm{dgap}_t = \varphi(b^t) + g(p^t, \beta^t).$$

For QL utilities, it is computed similarly, that is, through (32). For Leontief utilities, it is computed using the construction in Appendix D.3.

**Additional plots.** In §6, the plots for linear utilities are generated under random $B_i$. Here we present an augmented set of plots under different random market data, different utilities, unit v.s. varying, random budgets $B_i$ and different termination conditions ($\mathrm{dgap}_t/n \le \eta$ or $\epsilon(p^t, p^*) \le \eta$). All horizontal axes are market sizes $n$ (with $m = 2n$) and all vertical axes are number of iterations. The error vertical bars are the standard errors of the number of iterations across 30 repeats. The legends are in the subplot "Linear utilities, dgap/$n \le$1e-5".

Linear utilities, dgap/$n \le$ 1e-03

Linear utilities, dgap/$n \le$ 1e-04

Linear utilities, dgap/$n \le$ 1e-05

Linear utilities, dgap/$n \le$ 5e-06

QL utilities, $\varepsilon(p, p^*) \le$ 1e-02

QL utilities, $\varepsilon(p, p^*) \le$ 1e-03

QL utilities, dgap/$n$ ≤ 1e-03

Uniform   Normal   Exponential   Lognormal

QL utilities, dgap/$n$ ≤ 1e-04

Uniform   Normal   Exponential   Lognormal

QL utilities, dgap/$n$ ≤ 1e-05

Uniform   Normal   Exponential   Lognormal

QL utilities, dgap/$n$ ≤ 5e-06

Uniform   Normal   Exponential   Lognormal

QL utilities, dgap/$n$ ≤ 1e-03

Uniform   Normal   Exponential   Lognormal

QL utilities, dgap/$n$ ≤ 1e-04

Uniform   Normal   Exponential   Lognormal

QL utilities, dgap/$n \leq$ 1e-05

Uniform  Normal  Exponential  Lognormal

QL utilities, dgap/$n \leq$ 5e-06

Uniform  Normal  Exponential  Lognormal

Leontief utilities, $\varepsilon(u^t, u^*) \leq$ 1e-02

Uniform  Normal  Exponential  Lognormal

Leontief utilities, $\varepsilon(u^t, u^*) \leq$ 1e-03

Uniform  Normal  Exponential  Lognormal

Leontief utilities, dgap/$n \leq$ 1e-03

Uniform  Normal  Exponential  Lognormal

Leontief utilities, dgap/$n \leq$ 1e-04

Uniform  Normal  Exponential  Lognormal

Leontief utilities, dgap/$n \leq$ 1e-05

| Uniform | Normal | Exponential | Lognormal |
|---------|--------|-------------|-----------|

Leontief utilities, dgap/$n \leq$ 5e-06

| Uniform | Normal | Exponential | Lognormal |
|---------|--------|-------------|-----------|

[Supplementary Material 2 · supp.pdf]

## A  Preliminaries

We list and prove a few elementary lemmas used in subsequent proofs and discussions.

**Lemma 4** *Let $n \geq m$, $A \in \mathbb{R}^{m \times n}$ and $b \in \mathbb{R}^m$ be such that $\mathcal{S} = \{x \in \mathbb{R}^n : Ax = b\} \neq \emptyset$. Let the singular value decomposition (SVD) of $A^\top$ be $A^\top = U\Sigma V^\top$, where $U \in \mathbb{R}^{n \times r}$ and $V \in \mathbb{R}^{m \times r}$ have orthonormal columns, $\Sigma = \mathrm{Diag}(\sigma_1, \ldots, \sigma_r)$, $\sigma_1 \geq \ldots \sigma_r > 0$, $r = \mathrm{rank}(A)$. For any $x \in \mathbb{R}^n$, the projection of $x$ onto $\mathcal{S}$ can be expressed as*

$$\Pi_{\mathcal{S}}(x) = (I - UU^\top)x + U\Sigma^{-1}V^\top b.$$

*In particular, when $A$ has full rank,*

$$\Pi_{\mathcal{S}}(x) = (I - A^\top(AA^\top)^{-1}A)x + A^\top(AA^\top)^{-1}b.$$

*Proof.* The case of full rank $A$ is well-known, see, e.g., [54, Eq. (1)]. For general $A$, $Ax = b \Leftrightarrow V\Sigma U^\top x = b$. Recall that $U$ and $V$ consist of orthonormal columns and. Since $V\Sigma$ has full column rank, there exists a unique $w \in \mathbb{R}^r$ such that $V\Sigma w = b$. In fact, $w = \Sigma^{-1}V^\top b$. Therefore,

$$Ax = b \Leftrightarrow U^\top x = \Sigma^{-1}V^\top b.$$

Since $U^\top$ has full row rank, the formula follows directly from the full rank case. $\qquad\square$

**Lemma 5** *Under the same assumptions as Lemma 4, for any $x \in \mathbb{R}^n$, it holds that*

$$\sigma_r \cdot \|x - \Pi_{\mathcal{S}}(x)\| \leq \|Ax - b\|.$$

*Proof.* By Lemma (4) and the fact that $U$ and $V$ have orthonormal columns,

$$\|x - \Pi_{\mathcal{S}}(x)\| = \|U(U^\top x - \Sigma^{-1}V^\top b)\| = \|U^\top x - \Sigma^{-1}V^\top b\|$$
$$= \|\Sigma^{-1}V^\top(Ax - b)\| \leq \|\Sigma^{-1}\|\|Ax - b\|,$$

where $\|\Sigma^{-1}\| = \frac{1}{\sigma_r}$. $\qquad\square$

**Lemma 6** *For $B > 0$ and $g \in \mathbb{R}^d$, let*

$$x^* = \arg\min \left\{ \langle g, x \rangle + \sum_{i=1}^{d} x_i \log x_i : x \geq 0, \ \mathbf{1}^\top x = B \right\}.$$

*Then, $x_i^* = B \cdot \frac{e^{-g_i}}{\sum_\ell e^{-g_\ell}}$, $i \in [d]$.*

*Proof.* It can be easily verified via KKT optimality conditions. The Lagrangian is

$$L(x, \lambda) = \langle g, x \rangle + \sum_i x_i \log x_i - \lambda(\mathbf{1}^\top x - B).$$

By the first-order condition, for any $\lambda$, the minimizer $x(\lambda)$ of $L(x, \lambda)$ has $x_i(\lambda) = e^{\lambda - 1 - g_i}$. Primal feasibility implies $\sum_i x_i(\lambda) = B \Rightarrow e^\lambda = \frac{B}{\sum_i e^{-(1+g_i)}}$. Therefore,

$$x_i^* = B \cdot \frac{e^{-g_i}}{\sum_\ell e^{-g_\ell}}.$$

$\qquad\square$

## A.1 Proof of Theorem 1

Consider the minimization form of (1). Let $p_j \geq 0$ be the dual variable associated with constraint $\sum_i x_{ij} \leq 1$. The Lagrangian is

$$\mathcal{L}(x,p) = \left[ -\sum_i B_i \log u_i(x) + \left\langle p, \sum_i x_i \right\rangle - \sum_j p_j \right].$$

The Lagrangian dual is

$$\max_{p \geq 0} g(p) := \min_{x \geq 0} \mathcal{L}(x,p). \tag{8}$$

By assumption, there is a feasible $x$ to (1) with $u_i(x_i) > 0$ for all $i$. Note that $x' = \frac{1}{2}x$ is also feasible and $u_i(x_i') = \frac{1}{2}u_i(x_i) > 0$ by homogeneity of $u_i$. Consider $x'' = x' + \delta$, where $\delta > 0$ is sufficiently small so that $\sum_i x_{ij}'' < 1$ for all $j$. Then, $x'' > 0$ and $\sum_i x_{ij}'' < 1$ for all $j$. Therefore, (1) has a strictly feasible solution. Meanwhile, the dual (8) clearly has a strictly feasible solution $p > 0$ with $g(p) = \min_{x \geq 0} \mathcal{L}(x,p)$ finite. Therefore, strong duality holds by Slater's condition and the KKT conditions are necessary and sufficient for (primal and dual) optimality of a solution pair $(x,p)$ (see, e.g., [7, Appendix D]). Let $x^*$ and $p^*$ be optimal solutions to the primal (1) and dual (8), respectively. Clearly, $u_i(x_i^*) > 0$ for all $i$. By Lagrange duality, we have

$$x^* \in \arg\min_{x \geq 0} \mathcal{L}(x,p^*).$$

In other words, each $x_i^*$ maximizes $r_i(x_i, p^*) := B_i \log u_i(x_i) - \langle p^*, x_i \rangle$ on $x_i \geq 0$. We show that $(x^*, p^*)$ is a market equilibrium.

**Buyer optimality**  First, we verify that $\langle p^*, x_i^* \rangle = B_i$ for all $i$. Assume that $\langle p^*, x_i^* \rangle > B_i$ for some $i$. Let $\tilde{x}_i = (1 - \epsilon)x_i^*$, where $0 \leq \epsilon < 1$. Consider

$$\phi(\epsilon) = B_i \log u_i((1-\epsilon)x_i^*) - \langle p^*, (1-\epsilon)x_i^* \rangle = B_i \log u_i(x_i^*) - \langle p^*, x_i^* \rangle + B_i \log(1-\epsilon) + \epsilon\langle p^*, x_i^* \rangle.$$

Note that $\phi$ is differentiable on $(0,1)$ and $\phi'(\epsilon) = -\frac{B_i}{1-\epsilon} + \langle p^*, x_i^* \rangle$. Since $\langle p^*, x_i^* \rangle > B_i$, we have $\phi'(0) > 0$. In other words, replacing $x_i^*$ by $\tilde{x}_i$ with sufficiently small $\epsilon$ strictly decreases the value of $B_i \log u_i(x_i) - \langle p^*, x_i \rangle$, contradicting to the choice of $x^*$. Therefore, $\langle x^*, x_i^* \rangle \leq B_i$ for all $i$. Completely analogously, we can also show that $\langle p^*, x_i^* \rangle \geq B_i$ for all $i$. Therefore, for each buyer $i$, $x_i^*$ is feasible and depletes its budget $B_i$ under prices $p_i^*$. Hence, for any $x_i \in \mathbb{R}_+^m$, $\langle p^*, x_i \rangle \leq B_i$, since $x_i^*$ maximizes $r_i(\cdot, p^*)$, we have

$$B_i \log u_i(x_i^*) - \langle p^*, x_i^* \rangle \geq B_i \log u_i(x_i) - \langle p^*, x_i \rangle.$$

Since $\langle p^*, x_i \rangle \leq B_i = \langle p^*, x_i^* \rangle$, the above implies

$$B_i \log u_i(x_i^*) \geq B_i \log u_i(x_i).$$

Therefore, $u_i(x_i^*) \geq u_i(x_i)$. In other words, $x_i^* \in D_i(p^*)$ for all $i$.

**Market clearance**  By the complementary slackness condition in Lagrange duality, for item $j$ such that $\sum_i x_{ij}^* < 1$, it must holds that $p_j^* = 0$, completing the proof.

**Remark**  We can also assign any leftover of item $j$ to any buyer $i$ without violating its budget constraint, in order to "clear" the market. Meanwhile, since $u_i$ is CCNH, it is also "monotone" in the following sense: for any $\alpha \geq 0$,

$$u_i(x_i^* + \alpha \mathbf{e}^j) \geq u_i(x_i^*) + \alpha u_i(\mathbf{e}^j) \geq 0.$$

In other words, buyer $i$'s optimality is not affected by the assignment of any zero-price leftover.

## A.2 Characterizations of Hoffman constant

We compare our definition of Hoffman constant and another common, explicit characterization. Recall that $H_{\mathcal{X}}(A)$ is the smallest $H$ such that, for any $b$, $\mathcal{S} = \{x : Ax = b\}$,

$$\|x - \Pi_{\mathcal{X} \cap \mathcal{S}}(x)\| \leq H\|Ax - b\|, \; \forall\, x \in \mathcal{X}.$$

For any matrix $M$, let $\mathcal{B}(M)$ be the set of *nonsingular* submatrices consisting of rows of $M$. Define

$$H(M) = \max_{B \in \mathcal{B}(M)} \frac{1}{\sigma_{\min}(B)} < \infty. \tag{9}$$

The following fact is known (see, e.g., [34, §11.8] and [4, §2.1]).

**Lemma 7** *Suppose the reference polyhedral set can be represented by inequality constraints $\mathcal{X} = \{x : Cx \leq d\}$. Then,*

$$H_{\mathcal{X}}(A) \leq H\left(\begin{bmatrix} A \\ C \end{bmatrix}\right).$$

Clearly, $H(M)$ is finite for any $M$. In fact, this is the most well-known characterization of Hoffman constant, and is tight in the following sense: let $\mathcal{S} = \{x : Ax = b\}$ for some arbitrary right hand side $b$, then it is the smallest constant $H$ such that

$$\|x - \Pi_{\mathcal{X} \cap \mathcal{S}}(x)\| \leq H \left\| \begin{bmatrix} Ax - b \\ (Cx - d)_+ \end{bmatrix} \right\|$$

for all $x$ (not necessarily $\in \mathcal{X}$). However, for all of our purposes, that is, analysis of PG, $x$ is always restricted to be $\in \mathcal{X}$. Therefore, we choose to define $H_{\mathcal{X}}(A)$ as such, consistent with [5] and [53]. Meanwhile, the following is clear.

**Lemma 8** *For any matrices $A \in \mathbb{R}^{m \times n}$, $m \leq n$ and $C \in \mathbb{R}^{\ell \times n}$, it holds that*

$$H\left(\begin{bmatrix} A \\ C \end{bmatrix}\right) \geq \max\left\{\frac{1}{\sigma_{\min}(A)}, H(A)\right\}.$$

*Proof.* By definition (9), $H' := H\left(\begin{bmatrix} A \\ C^\top \end{bmatrix}\right) \geq H(A)$. If $\mathrm{rank}(A) = m$, then $H' \geq \frac{1}{\sigma_{\min}(A)}$

because $A \in \mathcal{B}\left(\begin{bmatrix} A \\ C \end{bmatrix}\right)$. If $r = \mathrm{rank}(A) < m$, let the (nonzero) singular values of $A$ be $\sigma_1 \geq \cdots \geq$

$\sigma_r = \sigma_{\min}(A) > 0$. Consider any $B \in \mathcal{B}(A) \subseteq \mathcal{B}\left(\begin{bmatrix} A \\ C \end{bmatrix}\right)$ with rank $r$ (having exactly $r$ rows), let

its nonzero singular values be $\sigma_1' \geq \cdots \geq \sigma_r' = \sigma_{\min}(B) > 0$. Applying Cauhchy's Interlacing Theorem (see, e.g., [31, Theorem 1]) on $AA^\top$ and its principal submatrix $BB^\top$, we have

$$\sigma_1 \geq \sigma_1' \geq \cdots \geq \sigma_r \geq \sigma_r'.$$

Therefore, $H' \geq \frac{1}{\sigma_{\min}(B)} \geq \frac{1}{\sigma_{\min}(A)}$. $\qquad\square$

### A.3 Proof of Theorem 2

We follow the development in [38, §4 & Appendix F] and further articulate the constants. There, the authors show that proximal gradient achieves linear convergence under the so-called *Proximal-PŁ* inequality. Consider the following general nonsmooth problem

$$F^* = \min_x F(x) = f(x) + g(x) \tag{10}$$

where $f$ is smooth convex with $L_f$-Lipschitz continuous gradient, $g$ is simple closed proper convex and $\mathrm{dom}\, g \subseteq \mathrm{dom}\, f$. One iteration of the proximal gradient method with stepsize $\gamma > 0$ is as follows:

$$x^{t+1} = \mathrm{Prox}_g\left(x^t - \gamma \nabla f(x^t)\right) = \arg\min_x \left[\langle \gamma \nabla f(x), x - x^t\rangle + \frac{1}{2}\|x - x^t\|^2 + g(x)\right]. \tag{11}$$

For any $\alpha > 0$ and any $x \in \mathrm{dom}\, g$, define

$$\mathcal{D}(x, \alpha) = -2\alpha \min_{x'} \left[\langle \nabla f(x), x' - x\rangle + \frac{\alpha}{2}\|x' - x\|^2 + g(x') - g(x)\right]. \tag{12}$$

Say that $F = f + g$ satisfies the proximal-PŁ inequality at $x$ w.r.t. $\Lambda \geq \lambda > 0$ if

$$\frac{1}{2}\mathcal{D}(x, \Lambda) \geq \lambda(F(x) - F^*), \tag{13}$$

Below is essentially [38, Theorem 5], which shows that the so-called Proximal-PŁ condition is sufficient for linear convergence. Note that, different from [38, Theorem 5], we only require (13) to hold for $x \in \mathcal{X}$ such that $F(x) \leq F(x^0)$ instead of all $x \in \mathcal{X}$. In addition, we note that in some cases (13) may hold with $\Lambda > L_f$, in which case the rate needs to be slightly adjusted. Since $\mathcal{D}(x, \cdot)$ is monotone [38, Lemma 1], (13) holds when $\Gamma$ is replaced by $\Gamma' \geq \Gamma$. The statement and proof are the same as [38, pp. 9] otherwise.

**Theorem 9** *Let $x^0 \in \operatorname{dom} g$. If $f$ and $g$ satisfies* (13) *for all $x \in \operatorname{dom} g$ such that $F(x) \leq F(x^0)$, then $x^t$ defined by* (11) *starting from $x^0$ with constant stepsize $\gamma = 1/L_f$ converges linearly with rate $1 - \frac{\lambda}{\bar{L}}$, where $\bar{L} = \max\{\Lambda, L_f\}$. In other words,*

$$F(x^t) - F^* \leq \left(1 - \frac{\lambda}{\bar{L}}\right)^t (F(x^0) - F^*), \ t = 1, 2, \dots$$

*Proof.* By assumption, (13) holds for all $x \in \operatorname{dom} g$, $x \leq F(x^0)$. In particular, it holds for $x^t$, $t, 1, 2, \dots$, since proximal gradient is a *descent* method, i.e., $F(x^0) \geq F(x^1) \geq \dots$ (see, e.g., [3, Corollary 10.18]). Therefore, by $L_f$-Lipschitz continuity of $\nabla f$, proximal gradient update (11), definition of $D(x, \cdot)$, its monotonicity, and (13) for all $x^t$,

$$\begin{aligned}
F(x^{t+1}) &\leq F(x^t) + \langle \nabla f(x^t), x^{t+1} - x^t \rangle + \frac{L_f}{2}\|x^{t+1} - x^t\|^2 + g(x^{t+1}) - g(x^t) \\
&\leq F(x^t) + \left[\langle \nabla f(x^t), x^{t+1} - x^t \rangle + \frac{\bar{L}}{2}\|x^{t+1} - x^t\|^2 + g(x^{t+1}) - g(x^t)\right] \\
&\leq F(x^t) - \frac{1}{2\bar{L}}\mathcal{D}(x^t, \bar{L}) \\
&\leq F(x^t) - \frac{\lambda}{\bar{L}}\left(F(x^t) - F^*\right) \\
\Rightarrow \ F(x^{t+1}) - F^* &\leq \left(1 - \frac{\lambda}{\bar{L}}\right)\left(F(x^t) - F^*\right).
\end{aligned}$$

Repeatedly applying the above inequality completes the proof. $\qquad\square$

Then, we prove Theorem 2. Clearly, problem (2) is (10) with $g(x) = \delta_{\mathcal{X}}(x)$ and PG is a special case of proximal gradient. By Theorem 9, in order to prove Theorem 2, it suffices to establish the Proximal-PŁ condition (13) (for all $x \in \mathcal{X}$, $f(x) \leq f(x^0)$ for some initial iterate $x^0$). Let $\mathcal{X}^*$ be the set of optimal solutions to (2) and $f^*$ be the optimal objective value. Since $h$ is $\mu$-strongly convex and $f(x) = h(Ax)$, there exists $z^* \in \operatorname{dom} f$ such that $\mathcal{S} = \{x : Ax = z^*\}$ and $\mathcal{X}^* = \mathcal{X} \cap \mathcal{S}$. Therefore, for any $x \in \mathcal{X}$, $x_p := \Pi_{\mathcal{X}^*}(x)$, we have

$$f(x_p) = h(Ax_p) \geq h(Ax) + \langle \nabla h(Ax), A(x_p - x)\rangle + \frac{\mu}{2}\|A(x_p - x)\|^2.$$

Note that

$$\langle \nabla h(Ax), A(x_p - x)\rangle = \langle A^\top \nabla h(Ax), x_p - x\rangle = \langle \nabla f(x), x_p - x\rangle.$$

Hence, for any $x \in \mathcal{X}$, by strong convexity of $h$ and definition of $H = H_{\mathcal{X}}(A)$, we have

$$\begin{aligned}
f(x_p) &\geq f(x) + \langle \nabla f(x), x_p - x\rangle + \frac{\mu}{2}\|A(x - x_p)\|^2 \\
&= f(x) + \langle \nabla f(x), x_p - x\rangle + \frac{\mu}{2}\|Ax - z^*\|^2 \\
&\geq f(x) + \langle \nabla f(x), x_p - x\rangle + \frac{\mu}{2H^2}\|x - x_p\|^2,
\end{aligned}$$

Therefore,

$$f^* \geq f(x) + \langle \nabla f(x), x_p - x \rangle + \frac{\mu}{2H^2} \|x - x_p\|^2$$

$$\geq f(x) + \min_{y \in \mathcal{X}} \left\{ \langle \nabla f(x), y - x \rangle + \frac{\mu}{2H^2} \|y - x\|^2 \right\}$$

$$\geq f(x) - \frac{H^2}{2\mu} \mathcal{D}\left(x, \frac{\mu}{H^2}\right)$$

$$\Rightarrow \frac{1}{2} \mathcal{D}\left(x, \frac{\mu}{H^2}\right) \geq \frac{\mu}{H^2}(f(x) - f^*).$$

Thus, (13) holds for all $x \in \mathcal{X}$, $f(x) \leq f(x^0)$ with

$$\Lambda = \lambda = \frac{\mu}{H^2}.$$

Since $\nabla f(x) = A^\top \nabla h(Ax)$ and $h$ is $(\mu, L)$-s.c., its Lipschitz constant can be chosen as

$$L_f = L\|A\|^2.$$

By Theorem 9, PG with stepsize $\gamma = \frac{1}{L_f}$ converges linearly with rate

$$1 - \frac{\frac{\mu}{H^2}}{\max\left\{\frac{\mu}{H^2}, L\|A\|^2\right\}} = 1 - \frac{\mu}{\max\{\mu, LH^2\|A\|^2\}}.$$

Finally, convergence of the distance to optimality $\|x^t - \Pi_{\mathcal{X}}(x^t)\|$ is straightforward: for any $x \in \mathcal{X}$, by the strong convexity of $h$ and definition of $H$,

$$f(x) - f^* = h(Ax) - h(Ax_p) \geq \frac{\mu}{2}\|Ax - Ax_p\|^2 = \frac{\mu}{2}\|Ax - z^*\|^2 \geq \frac{\mu}{2H}\|x - x_p\|^2.$$

□

**Remark** A special case is when $d \geq r$ (recall that $A \in \mathbb{R}^{d \times r}$) and $\mathrm{rank}(A) = r$. In this case, $f(x) = h(Ax)$ itself is strongly convex with modulus $\mu \sigma_{\min}(A)^2$. In this case, classical analysis (e.g., [3, §10.6]) implies linear convergence with rate $1 - \frac{\mu \sigma_{\min}(A)^2}{L\|A\|^2}$. Meanwhile, in the above analysis, we have $\mathcal{X}^* = \{x^*\} = \mathcal{S} = \{x : Ax = z^*\} = \mathcal{X} \cap \mathcal{S}$ (since $x^*, z^*$ are unique and $\mathrm{rank}(A) = r$). By Lemma 5, for any $x$, it holds that

$$\|x - \Pi_{\mathcal{X}^*}(x)\| \leq \frac{1}{\sigma_{\min}(A)^2}\|Ax - z^*\|.$$

Therefore, by the definition of Hoffman constant, $H_{\mathcal{X}}(A) \leq \frac{1}{\sigma_{\min}(A)^2}$ and the classical rate under strong convexity is recovered.

## A.4 Proof of Theorem 3

Let $\mathcal{X}^*$ be the set of optimal solutions to (2). First, recall the following lemma [64, Lemma 14], which ensures the first part of the theorem, that is, uniqueness of $Ax^*$ and $q^\top x^*$ for all $x^* \in \mathcal{X}^*$.

**Lemma 9** *There exist unique $z^* \in \mathbb{R}^r$ and $w^* \in \mathbb{R}$ such that for any $x^* \in \mathcal{X}^*$,*

$$Ax^* = z^*, \; \langle q^*, x \rangle = w^*.$$

The next lemma is essentially [4, Lemma 2.5]. Different from the statement of [4, Lemma 2.5], we keep $\|\nabla h(z^*)\|$ instead of bounding it by $\sup_{x \in \mathcal{X}} \|\nabla h(Ax)\|$. We also define $C = f(x^0) - f^*$ instead of $C = \sup_{x \in \mathcal{X}} f(x) - f^*$, since subsequent application of the lemma only involves PG iterates $x^t$, which have monotone decreasing objective values $f(x^0) \geq f(x^1) \geq \ldots$ The proof remains unchanged otherwise.

**Lemma 10** *Let $z^*$ be as in Lemma 9 and $x^0 \in \mathcal{X}$. For any $x \in \mathcal{X}$ such that $f(x) \leq f(x^0)$, it holds that*

$$\|x - \Pi_{\mathcal{X}^*}(x)\|^2 \leq \kappa \left( f(x) - f^* \right),$$

*where, same as in Theorem 3, $\kappa = H_{\mathcal{X}}(A)^2 \left( C + 2GD_A + \frac{2(G^2+1)}{\mu} \right)$, $C = f(x^0) - f^*$,*
*$G = \|\nabla h(z^*)\|$, $D_A = \sup_{x,y \in \mathcal{X}} \|A(x-y)\|$.*

Finally, take $L_f = L\|A\|^2$ as a Lipschitz constant of $\nabla f$. By Lemma 10 and [38, §4.1], it holds that (2) satisfies the proximal-PŁ inequality (13) with

$$\Lambda = \lambda = \frac{1}{2\kappa}$$

for all $x \in \mathcal{X}$ such that $f(x) \leq f(x^0)$ (in particular, for all $x^t$, $t = 1, 2, \dots$). By Theorem 9, PG converges linearly with rate $1 - \frac{\lambda}{\max\{\Lambda, L_f\}} = 1 - \frac{1}{\max\{1, 2\kappa L\|A\|^2\}}$.

**Remark** Lemma 10 shows that QG holds. Similar convergence guarantees can also be derived from other QG-based analysis, e.g., [27, Corollary 3.7].

### A.5 Linear convergence of PG with linesearch

First, we consider the more general proximal gradient setup (10). Let $L_f$ be a Lipschitz constant of $\nabla f$ and the Proximal-PŁ inequality 13 holds with $\Lambda \geq \lambda \geq 0$ for all $x \in \operatorname{dom} g$ such that $F(x) \leq F(x^0)$. Let $\alpha \geq 1$, $\beta \in (0,1)$, $\Gamma > 0$ (increment factor, decrement factor, upper bound on stepsize, respectively). The linesearch subroutine $\mathcal{LS}_{\alpha,\beta,\Gamma}$ is defined in Algorithm 1.

---

**Algorithm 1** $x_{t+1}, \gamma_t, k_t \leftarrow \mathcal{LS}_{\alpha,\beta,\Gamma}(x, \gamma, k_{\text{prev}})$ with parameters $\alpha \geq 1$, $\beta \in (0,1)$, $\Gamma > 0$.

---

If $k_{\text{prev}} = 0$, set $\gamma^{(0)} = \min\{\alpha\gamma, \Gamma\}$. Otherwise, set $\gamma^{(0)} = \gamma$.
For $k = 0, 1, 2, \dots$
    1. Compute $x^{(k)} = \operatorname{Prox}_{\lambda^{(k)}g}(x - \gamma^{(k)}\nabla f(x))$.
    2. Break if

$$f(x^{(k)}) \leq f(x) + \langle \nabla f(x), x^{(k)} - x \rangle + \frac{1}{2\gamma^{(k)}}\|x^{(k)} - x\|^2. \qquad (14)$$

    3. Set $\gamma^{(k+1)} = \beta\gamma^{(k)}$ and continue to $k + 1$.
Return $x_{t+1} = x^{(k)}$, $\gamma_t = \gamma^{(k)}$, $k_t = k$.

---

In this way, proximal gradient with linesearch can be described formally as follows: starting from $x^0 \in \operatorname{dom} f$, $\gamma_{-1} = \Gamma$, $k_{-1} = 0$, perform the following iterations

$$(x^{t+1}, \gamma_t, k_t) \leftarrow \mathcal{LS}_{\alpha,\beta,\Gamma}(x^t, \gamma_{t-1}, k_{t-1}), \quad t = 1, 2, \dots$$

Note that (14) holds for any $\gamma^{(k)} \leq \frac{1}{L_f}$ (see, e.g., [3, Theorem 10.16]). Therefore, Algorithm 1 terminates when $\gamma^{(0)}\beta^k \leq \frac{1}{L_f}$. This means

$$\gamma_t \geq \tilde{\gamma} := \min \left\{ \Gamma, \frac{\beta}{L_f} \right\}. \qquad (15)$$

for all $t$. Note that we explicitly include the case of $\Gamma \leq \frac{1}{L_f}$, although in practice $\Gamma$ is often set very large. Clearly,

$$\Gamma\beta^k \leq \tilde{\gamma} \iff k \geq \frac{\log \frac{\Gamma}{\tilde{\gamma}}}{\log \frac{1}{\beta}}.$$

Therefore, in Algorithm 1, the backtracking iteration index satisfies $k_t \leq \frac{\log \frac{\Gamma}{\tilde{\gamma}}}{\log \frac{1}{\beta}}$ for all $t$. Note that if the loop breaks at $k_t$, the number of $\operatorname{Prox}$ evaluations is exactly $k_t + 1$.

659 Let

$$\bar{L} = \max\left\{\frac{1}{\tilde{\gamma}}, \Lambda\right\} = \max\left\{\frac{1}{\Gamma}, \frac{L_f}{\beta}, \Lambda\right\}. \tag{16}$$

660 Then, monotonicity of $D(x, \cdot)$ implies, for all $x \in \text{dom } g$ such that $F(x) \leq F(x^0)$,

$$\frac{1}{2}\mathcal{D}(x, \bar{L}) \geq \frac{1}{2}\mathcal{D}(x, \Lambda) \geq \lambda\left(F(x) - F^*\right).$$

661 Following the proof of Theorem 9 (or that of [38, Theorem 5]), we have

$$F(x^{t+1}) \leq F(x^t) + \langle \nabla f(x^t), x^{t+1} - x^t \rangle + \frac{L_f}{2}\|x^{t+1} - x^t\|^2 + g(x^{t+1}) - g(x^t)$$

$$\leq F(x^t) + \langle \nabla f(x^t), x^{t+1} - x^t \rangle + \frac{\bar{L}}{2}\|x^{t+1} - x^t\|^2 + g(x^{t+1}) - g(x^t)$$

$$\leq F(x^t) - \frac{1}{2\bar{L}}\mathcal{D}\left(x^t, \bar{L}\right)$$

$$\leq F(x^t) - \frac{\lambda}{\bar{L}}(F(x^t) - F^*)$$

$$\Rightarrow F(x^{t+1}) - F^* \leq \left(1 - \frac{\lambda}{\bar{L}}\right)\left(F(x^t) - F^*\right).$$

662 Summarizing the above discussion, we have the following convergence guarantee for PG with
663 linesearch.

664 **Theorem 10** *Let $\alpha \geq 1$, $\beta \in (0, 1)$ and $\Gamma > 0$. For problem* (10) *satisfying the Proximal-*
665 *PŁinequality with $\Lambda \geq \lambda > 0$ for all $x \in \text{dom } g$ such that $F(x) \leq F(x^0)$, proximal gradient*
666 (11) *with linesearch subroutine $\mathcal{LS}_{\alpha,\beta,\Gamma}$ described in Algorithm 1 generates iterates $x^t$ such that*

$$F(x^{t+1}) - F^* \leq \left(1 - \frac{\lambda}{\bar{L}}\right)^t \left(F(x^0) - F^*\right), \quad t = 1, 2, \ldots, \tag{17}$$

667 *where $\bar{L}$ is defined in* (16). *Furthermore, each iteration requires at most $1 + \frac{\log \frac{\Gamma}{\tilde{\gamma}}}{\log \frac{1}{\beta}}$ number of $\text{Prox}$*
668 *evaluations.*

669 *Proof of Theorem 4.* In the above discussion, when $g(x) = \delta_{\mathcal{X}}(x)$, we can replace the Lipschitz
670 constant $L_f$ by the restricted one $L_f^{\mathcal{X}}$ throughout, since Algorithm 1 ensures $x^t \in \mathcal{X}$ for all $t$. It
671 remains to apply Theorem 10. For $q = 0$, $\Lambda = \lambda = \frac{\mu}{H^2}$ and $\bar{L} = \max\left\{\frac{1}{\Gamma}, \frac{L_f^{\mathcal{X}}}{\beta}, \frac{\mu}{H^2}\right\}$. Therefore, the
672 rate is

$$1 - \frac{\lambda}{\bar{L}} = 1 - \frac{\mu}{\max\{\mu, H^2/\Gamma, H^2 L_f^{\mathcal{X}}/\beta\}}.$$

673 For $q \neq 0$, $\Lambda = \lambda = \frac{1}{2\kappa}$ and $\bar{L} = \max\left\{\frac{1}{\Gamma}, \frac{L_f^{\mathcal{X}}}{\beta}, \frac{1}{2\kappa}\right\}$. Therefore, the rate is

$$1 - \frac{1}{\max\{1, 2\kappa L_f^{\mathcal{X}}/\beta, 2\kappa/\Gamma\}}.$$

674 $\qquad\qquad\qquad\qquad\qquad\qquad\qquad\qquad\qquad\qquad\qquad\qquad\qquad\qquad\qquad\qquad\qquad\square$

## A.6  Other utility functions

676 Recall that, by Theorem (1), for any CCNH utilities $u_i$, optimal solutions to the EG convex program
677 (1) correspond to equilibrium allocation and prices.

678 **CES utilities** are parametrized by a nondegenerate $v$ and exponent $\rho \in (-\infty, 1]\backslash\{0\}$:

$$u_i(x_i) = \left(\sum_{j=1}^m v_{ij} x_{ij}^\rho\right)^{1/\rho}.$$

Clearly, $\rho = 1$ gives linear utilities. For $\rho < 1$, it has been shown that Proportional Response dynamics achieves linear convergence in prices and utilities [67, Theorem 4] under their notion of $\epsilon$-approximate market equilibrium [67, pp. 2693].

**Cobb-Douglas utilities** represent substitutive items and take the following form, for parameters $\lambda = (\lambda_i)$, $\lambda_i \in \Delta_m$:

$$u_i(x_i) = \Pi_j x_{ij}^{\lambda_{ij}}.$$

In this case, EG (1) decomposes item-wise into simple problems with explicit solutions. Specifically, for each item $j$, the minimization problem is

$$\min_{x_{:,j} \in \Delta_n} - \sum_i B_i \lambda_{ij} \log x_{ij}.$$

Let $p_j$ be the Lagrangian multiplier associated with constraint $\sum_i x_{ij} = 1$. The Lagrangian is

$$\mathcal{L}(x_{:,j}, p_j) = - \sum_i B_i \lambda_{ij} \log x_{ij} + p_j \left( \sum_i x_{ij} - 1 \right).$$

By first-order stationarity condition, for any $p_j \in \mathbb{R}$, $\mathcal{L}(x_{:,j}, p_j)$ is minimized when

$$x_{ij} = \frac{B_i \lambda_{ij}}{p_j}. \tag{18}$$

Substituting it into $\mathcal{L}$ and discarding the constants w.r.t. $p_j$, we have

$$g(p_j) = \left( \sum_i B_i \lambda_{ij} \right) \log p_j - p_j,$$

which is maximized at equilibrium prices

$$p_j^* = \sum_i B_i \lambda_{ij}.$$

Therefore, by 18, the equilibrium $x^*$ under Cobb-Douglas utilities is given by

$$x_{ij}^* = \frac{B_i \lambda_{ij}}{\sum_i B_i \lambda_{ij}}, \ \forall i, j.$$

# B  Linear utilities

## B.1  Shmyrev's convex program

Under linear utilities, it turns out that we can also compute market equilibrium via the following convex program due to Shmyrev [58, 8]. In this convex program, the variables are the *bids* $b_{ij}$, $i \in [n]$, $j \in [m]$ and prices $p_j$, $j \in [m]$.

$$\max \sum_{i,j} b_{ij} \log v_{ij} - \sum_j p_j \log p_j \ \text{s.t.} \ \sum_i b_{ij} = p_j, \ j \in [m], \ \sum_j b_{ij} = B_i, \ i \in [n], \ b \geq 0. \tag{19}$$

Given an optimal solution $b^*$, equilibrium prices and allocations are then given by $p_j^* = \sum_i b_{ij}^*$ and $x_{ij}^* = \frac{b_{ij}^*}{p_j^*}$, respectively.

## B.2  Proof of Lemma 1

Any $x \in \mathcal{X}$ satisfies $x \leq 1$. Therefore, $\langle v_i, x_i \rangle \leq \|v_i\|_1 \|x_i^*\|_\infty \leq \|v_i\|_1 = \bar{u}_i$. For the lower bound, recall that at an equilibrium allocation $x^*$ ensures that every buyer gets at least the utility of the proportional share, that is,

$$\langle v_i, x_i^* \rangle \geq \left\langle v_i, \frac{B_i}{\|B\|_1} \mathbf{1} \right\rangle = \frac{B_i \|v_i\|_1}{\|B\|_1} = \underline{u}_i.$$

 **B.3   Uniqueness of equilibrium quantities and convergence of $u^t$, $p^t$**

Convergence of $u^t$ to $u^*$ can be easily seen as follows. Let $x^t$ be the PG iterates and $\tilde{h}$, $A$, $f = \tilde{h}(Ax)$, $\mu$ be defined as in §3 and $f^* = \min_{x \in \mathcal{X}} f(x)$. Since $\tilde{h}$ is $\mu$-strongly convex, we have

$$\frac{\mu}{2}\|u^t - u^*\|^2 \le \tilde{h}(u^t) - \tilde{h}(u^*) \le \tilde{h}(Ax^t) - f^*,$$

which converges linearly. Next, we show uniqueness of $p^*$ via simple arguments and construct a sequence of linearly convergent prices $p^t$.

**Lemma 11** *Assume that $v$ is nondegenerate. Then, the equilibrium prices $p^*$ under linear utilities are unique.*

*Proof.* By Theorem 1 and [21, Lemma 3], $p^*$ is an optimal solution (together with some $\beta^*$) to the following problem (dual of (1) with linear utilities): □

$$\min_{p, \beta} \sum_j p_j - \sum_i B_i \log \beta_i \ \text{ s.t. } p \ge 0, \ \beta \ge 0, \ p_j \ge v_{ij}\beta_i, \ \forall\, i, j. \tag{20}$$

Here, strong duality holds since there clearly exist primal and dual strictly feasible solutions with finite objective values given nondegenerate $v$ (c.f. Theorem 1 and Appendix A.1). We can eliminate $p$ by letting $p_j = \max_i v_{ij}\beta_i$ for all $j$ and rewrite (20) as

$$\min_{\beta} \sum_j \max_i v_{ij}\beta_j - \sum_i B_i \log \beta_i \ \text{ s.t. } \beta \ge 0.$$

In the above, since the objective is strongly convex and the feasible region is $\beta \ge 0$, the optimal solution $\beta^*$ is clearly unique. Furthermore, it must hold that $\beta^* > 0$ (since the optimal objective value is finite and strong duality holds). For $p^*$ optimal to (20), it must hold that $p_j^* = \max_i v_{ij}\beta_i^*$. In fact, $p_j^* \ge \max_i v_{ij}\beta_i^*$ by feasibility and, for any strict inequality, decreasing the corresponding $p_j^*$ strictly decreases the objective. □

The following lemma provides simple upper and lower bounds on feasible and equilibrium prices, respectively. The lower bounds are slightly strengthened over the existing one [8, Lemma 17].

**Lemma 12** *Let $p^*$ be equilibrium prices under linear utilities with nondegenerate valuations $v$. Then, $\underline{p}_j \le p_j^* \le \bar{p}_j$ for all $j$, where $\underline{p}_j = \max_i \frac{v_{ij}B_i}{\|v_i\|_1}$ and $\bar{p}_j = \|B\|_1$.*

*Proof.* It is essentially the same as the proof of Lemma 2, except that, at optimality, $u_i \le \|v_i\|_1 + B_i$ can be strengthened to $u_i \le \|v_i\|_1$ (utility of each buyer is at most that of having a unit of every item). □

**A linearly convergent sequence of $p^t$**   Here, all norms are vector norms. Note that each step of PG is of the form $x^{t+1} = \Pi_{\mathcal{X}}(\bar{x}^t)$, where $\bar{x}^t = x^t - \gamma \nabla f(x^t)$. Since $\nabla f$ is $L_f$-Lipschitz, the mapping

$$\phi_1 : x \mapsto x - \gamma \nabla f(x^t)$$

is Lipschitz continuous (w.r.t. $\|\cdot\|_2$) with constant $1 + \gamma L_f = 2$ (where $\gamma = \frac{1}{L\|A\|^2}$ is the fixed stepsize). Meanwhile, we have the following.

**Lemma 13** *Let $y \in \mathbb{R}^n$ and $y^* = \Pi_{\Delta^n}(y)$. There exists a unique multiplier $\lambda \in \mathbb{R}$, which can be computed in $O(n \log n)$ time, such that*

$$\sum_{i=1}^{n} (y_i - \lambda)_+ = 1. \tag{21}$$

*Moreover, the mapping $\phi_2 : y \mapsto \lambda$ is piecewise linear and 1-Lipschitz continuous w.r.t. $\|\cdot\|_1$.*

*Proof.* By the KKT conditions for simplex projection (see, e.g., [65, §3]), it holds that there exists unique $\lambda$ such that

$$y^* = (y - \lambda\mathbf{1})_+.$$

Suppose there exists $\lambda_1 < \lambda_2$ that satisfy (21). Then, since the left-hand side of (21), denoted as $w(\lambda)$, is monotone decreasing in $\lambda$, it must hold that $w(\lambda) = 1$ for all $\lambda \in [\lambda_1, \lambda_2]$. In other words, $w(\cdot)$ is *constant* on $[w_1, w_2]$. This further implies $w(\lambda) = 0$ for all $w \in [w_1, w_2]$, a contradiction. Therefore, $\lambda = \phi_2(y)$ is uniquely defined. Let $I^+(y)$, $I^0(y)$, $I^-(y)$ denote the set of indices $i \in [n]$ such that $y_i > \lambda$, $y_i = \lambda$, $y_i < \lambda$, respectively (where $\lambda = \phi_2(y)$). We have

$$\lambda = \frac{\sum_{i \in I^+(y)} y_i - 1}{|I^+(y)|} = \frac{\sum_{i \in I^+(y) \cup I^0(y)} y_i - 1}{|I^+(y)| + |I^0(y)|},$$

which is piecewise linear in $y$ since there are only finitely many index possible sets of indices and $I^+(y)$ is always nonempty (otherwise $\sum_i (y_i - \lambda)_+ = 0$). To see Lipschitz continuity, let $y'$ be such that $\|y' - y\|_1 \le \epsilon$, where $0 < \epsilon < \min\{|y_i - y_j| : i, j \in [n], y_i \ne y_j\}$. It must hold that $I^+(y) \subseteq I^+(y')$. In other words, $\lambda' = \phi_2(y')$ does not deactivate any $i \in I^+(y)$, only bringing new $i \in I^0(y)$. Hence, it holds that $|\lambda' - \lambda| \le \frac{\|y - y'\|_1}{|I^+(y)|} \le \|y - y'\|_1$. In other words, $\phi_2$ is 1-Lipschitz continuous w.r.t. $\|\cdot\|_1$.

Finally, [65, Algorithm 1]) computes $\lambda$ and $y^*$ in $O(n \log n)$ time. $\qquad\square$

Slightly abusing the notation, let $\phi_2$ also denote the mapping from $x \in \mathbb{R}^{n \times m}$ to $\lambda \in \mathbb{R}^m$, that is, $\lambda_j = \varphi_2(x_{1j}, \ldots, x_{nj})$. Let
$$\phi(x) = \phi_2(\phi_1(x))/\gamma$$
and $p^t = \phi(x^t)$. Here, $\phi_1$ is 2-Lipschitz continuous and $\phi_2$ is 1-Lipschitz continuous w.r.t. $\|\cdot\|_1$. For any optimal solution $x^* \in \mathcal{X}^*$, by $x^* = \Pi_{\mathcal{X}}(x^*)$ and KKT conditions for (1) and (20), it can be seen that
$$p^* = \phi(x^*).$$

Using the Lipschitz continuity properties of $\phi_1, \phi_2$ and Theorem 2 (properties ), we have

$$\|p^t - p^*\|_1 = \|\phi(x^t) - \phi(\Pi_{\mathcal{X}^*}(x^t))\|_1 \le \frac{1}{\gamma}\|\phi_1(x^t) - \phi_1(\Pi_{\mathcal{X}^*}(x^t))\|_1$$

$$\le \frac{n}{\gamma}\|\phi_1(x^t) - \phi_1(\Pi_{\mathcal{X}^*}(x^t))\| \le \frac{2n}{\gamma} \cdot \|x^t - \Pi_{\mathcal{X}^*}(x^t)\|$$

$$\le \frac{2n}{\gamma} \cdot \sqrt{\frac{2H_{\mathcal{X}}(A)}{\mu}\left(f(x^t) - f^*\right)}$$

$$\le \frac{2n}{\gamma}\sqrt{\frac{2H_{\mathcal{X}}(A)}{\mu}} \cdot \left(1 - \frac{\mu}{2HL\|A\|^2}\right)^{t/2} \cdot \sqrt{f(x^0) - f^*}.$$

Therefore, we can take $C = \frac{2n}{\gamma}\sqrt{\frac{2H_{\mathcal{X}}(A)}{\mu}} \cdot \sqrt{f(x^0) - f^*}$ and $\rho = \sqrt{1 - \frac{\mu}{2HL\|A\|^2}} \in (0, 1)$.

Since $p^* \ge \underline{p} > 0$, we can bound the maximum relative price error $\eta^t = \max_j \frac{|p_j^t - p_j^*|}{p_j^*}$ as follows, where $\underline{p}_{\min} = \min_j \underline{p}_j$.

$$\eta^t \le \frac{\|p^t - p^*\|_1}{\underline{p}_{\min}} \le \frac{C}{\underline{p}_{\min}} \cdot \rho^t.$$

# C  QL utilities

## C.1  Derivation of the QL-Shmyrev convex program (4)

In [21, Lemma 5], the convex program for the equilibrium prices is as follows:

$$\min \sum_j p_j - \sum_i B_i \log \beta_i \text{ s.t. } v_{ij}\beta_i \le p_j, \, \forall i, j, \ 0 \le \beta \le 1. \tag{22}$$

Note that it is simply the dual of EG under linear utilities (20) with additional constraints $\beta \le 1$. Assuming $v$ is nondegenerate, by a change of variable and Lagrange duality, we can derive the dual

of (22). First, at optimality, it must holds that $\beta_i > 0$ for all $i$. Therefore, by nondegeneracy of $v$, $p_j > 0$ for all $j$ at optimality. Let $p_j = e^{q_j}$ and $\beta_i = e^{-\gamma_i}$. The above problem is equivalent to

$$
\begin{aligned}
\min \quad & \sum_j e^{q_j} + \sum_i B_i \gamma_i \\
\text{s.t.} \quad & q_j + \gamma_i \geq \log v_{ij}, \ \forall\, i, j, \\
& \gamma \geq 0.
\end{aligned}
\tag{23}
$$

Let $b_{ij} \geq 0$ be the dual variable associated with constraint $q_j + \gamma_i \geq \log v_{ij}$. The Lagrangian is

$$
\begin{aligned}
L(q, \gamma, b) &:= \sum_j e^{q_j} + \sum_i B_i \gamma_i - \sum_{i,j} b_{ij} \left( q_j + \gamma_i - \log v_{ij} \right) \\
&= \sum_j \left( e^{q_j} - \left( \sum_i b_{ij} \right) q_j \right) + \sum_i (B_i - \sum_j b_{ij}) \gamma_i + \sum_{i,j} (\log v_{ij}) b_{ij}.
\end{aligned}
$$

Clearly, when $\sum_j b_{ij} \leq B_i$ for all $i$, $\gamma \geq 0$, $L(q, \gamma, b)$ is minimized at $q_j = \log \sum_i b_{ij}$ and $\gamma = 0$. When $\sum_j b_{ij} > B_i$ for some $i$, $L \to -\infty$ as $\gamma_i \to \infty$. Therefore, when $\sum_j b_{ij} \leq B_i$ for all $i$, we have

$$
g(b) = \sum_j \left[ \sum_{i,j} b_{ij} - \left( \sum_i b_{ij} \right) \log \sum_i b_{ij} \right] + \sum_{i,j} (\log v_{ij}) b_{ij}.
$$

Therefore, the dual is

$$
\max g(b) \ \text{s.t.} \ b \geq 0, \ \sum_j b_{ij} \leq B_i, \ \forall\, i.
$$

Adding slack variables $\delta = (\delta_1 \ldots, \delta_n)$ and writing it in minimization form yield (4).

**Remark** When some $v_{ij} = 0$ (but $v$ is still nondegenerate), by the above derivation, the first summation in (4) should be replaced by $\sum_{(i,j) \in \mathcal{E}}$, where $\mathcal{E} = \{(i,j) : v_{ij} > 0\}$. The dual remains the same otherwise.

## C.2 Proof of Lemma 2

Similar to the proof of Lemma 11, this can be seen via the uniqueness of the optimal solution $(p^*, \beta^*)$ of (22), that is, from uniqueness of $\beta^*$ to that of $p_j^* = \max_i v_{ij} \beta_i^*$.

Let $(b^*, \delta^*)$ be an optimal solution to (4). Note that strong duality holds for (23) and (4), since there exit simple strictly feasible solutions. By the derivation in Appendix C.1, it holds that $q_j^* = \log \sum_i b_{ij}^*$ gives an optimal solution to (23) (the first-order optimality condition). Therefore,

$$
p_j^* = e^{q_j^*} = \sum_i b_{ij}^*.
$$

Next we establish the upper and lower bounds on $p^*$. By the derivation in Appendix C.1 and Lagrange duality, for any optimal solution $b^*$ to (4), it holds that $p_j^* := \sum_i b_{ij}^*$ and $\beta_j^* = \min_{j \in J_i} \frac{p_j^*}{v_{ij}}$ give the (unique) optimal solution to (22). Clearly, $\beta^* \leq 1$ and therefore

$$
p_j^* = \max_i v_{ij} \beta_i^* \leq \max_i v_{ij} = \bar{p}_j.
$$

By [21, Lemma 5], the dual of (22) is (c.f. the original EG primal 1)

$$
\begin{aligned}
\max_{u,\, x,\, s} \quad & \sum_i B_i \log u_i - s_i \\
\text{s.t.} \quad & u_i \leq v_i^\top x_i + s_i, \ \forall\, i, \\
& \sum_i x_{ij} \leq 1, \ \forall\, j, \\
& x, s \geq 0.
\end{aligned}
\tag{24}
$$

Clearly, strong duality holds for (22) and (24). Furthermore, notice the following.

783 • $\beta_i^* = \frac{B_i}{u_i^*}$ at optimality, where $u_i^*$ is the amount of utility of buyer $i$. This is by the stationarity
784 condition in the KKT optimality conditions.

785 • $u_i^* \leq \|v_i\|_1 + B_i$, where the right hand side is the amount of utility of all items and the entire
786 budget. This can also be seen as follows. When $s_i > B_i$, decreasing $s_i$ strictly increases the
787 objective of (24). Therefore, the optimal $s^*$ must satisfy $s_i^* \leq B_i$. It then follows from the
788 constraint $u_i \leq v_i^\top x_i + s_i$.

789 Therefore,

$$p_j^* \geq \max_i v_{ij} \beta_i^* \geq \max_i \frac{v_{ij} B_i}{\|v_i\|_1 + B_i} = \underline{p}_j.$$

790 ## C.3 Proof of Theorem 7

791 Similar to [8, Lemma 7], we first establish the following "generalized Lipschitz condition" for $\varphi$,
792 which is key to the claimed last-iterate convergence.

793 **Lemma 14** *For all* $(b, \delta), (b', \delta') \in \mathcal{B}$, *it holds that*

$$\varphi(b') \leq \varphi(b) + \langle \nabla \varphi(b), b' - b \rangle + D(b', \delta' \| b, \delta). \tag{25}$$

794

795 *Proof.* Recall that $p_j(b) = \sum_i b_{ij}$, $\frac{\partial}{\partial b_{ij}} \varphi(b) = \log \frac{p_j(b)}{v_{ij}}$. For $(a, \delta^a), (b, \delta^b) \in \mathcal{B}$, we have

$$\begin{aligned}
\varphi(b) &- \varphi(a) - \langle \nabla \varphi(a), b - a \rangle \\
&= -\sum_{i,j}(1 + \log v_{ij})(b_{ij} - a_{ij}) + \sum_j p_j(b) \log p_j(b) - \sum_j p_j(a) \log p_j(a) \\
&\quad - \sum_{i,j}(b_{ij} - a_{ij}) \log \frac{p_j(a)}{v_{ij}} \\
&= -\sum_{i,j}(b_{ij} - a_{ij}) + \sum_j p_j(b) \log \frac{p_j(b)}{p_j(a)} \\
&= \sum_i(\delta_i^b - \delta_i^a) + \sum_j p_j(b) \log \frac{p_j(b)}{p_j(a)}. \tag{26}
\end{aligned}$$

796 Note that convexity and smoothness of $x \mapsto x \log \frac{x}{y}$ $(y > 0)$ implies

$$\delta_i^b - \delta_i^a \leq \delta_i^b \log \frac{\delta_i^b}{\delta_i^a}. \tag{27}$$

797 Meanwhile, as in the proof of [8, Lemma 7], by convexity of $q(x, y) = x \log \frac{x}{y}$, it holds that

$$\sum_j p_j(b) \log \frac{p_j(b)}{p_j(a)} \leq \sum_{i,j} b_{ij} \log \frac{b_{ij}}{a_{ij}}. \tag{28}$$

798 By (27) and (28), the right hand side of (26) can be bounded by $D(b, \delta^b \| a, \delta^a)$. Therefore, (25) holds.
799 □

800 Next, we prove the inequality on the right. Clearly, $(b^0, \delta^0) \in \mathcal{B}$. By [8, Theorem 3] (with objective
801 $f = \varphi$, constraint set $C = \mathcal{B}$ and stepsize $\gamma$), we have

$$\varphi(b^t) - \varphi(b^*) \leq \frac{D(b^*, \delta^* \| b^0, \delta^0)}{t}.$$

802 Similar to the proof of [8, Lemma 13], we can bound the Bregman divergence on the right hand side
803 as follows, where $b_{ij} = \delta_i = \frac{B_i}{m+1}$.

$$\begin{aligned}
D(b^*, \delta^* \| b^0, \delta^0) &= \sum_{i,j} b_{ij}^* \log \frac{b_{ij}^*}{B_i} + \sum_i \delta_i^* \log \frac{\delta_i^*}{B_i} + \sum_{i,j} b_{ij}^* \log(m+1) + \sum_i \delta_i^* \log(m+1) \\
&\leq \sum_{i,j} b_{ij}^* \log(m+1) + \sum_i \delta_i^* \log(m+1) \\
&\leq \|B\|_1 \log(m+1),
\end{aligned}$$

where the first inequality is because $\frac{b_{ij}^*}{B_i} \le 1$. Combining the above yields the desired inequality.[1]

Finally, we show the inequality on the left. By optimality of $(b^*, \delta^*)$, we have

$$\langle \nabla\varphi(b^*), b - b^* \rangle \ge 0, \ \ \forall (b, \delta) \in \mathcal{B}.$$

Recall that $p_j(b) = \sum_i b_{ij}$. By (26), we have

$$D(p^t \| p^*) = -\sum_{i,j}(b_{ij}^t - b_{ij}^*) + \sum_j p_j(b^t)\log \frac{p_j(b^t)}{p_j(b^*)} \le \varphi(b^t) - \varphi^*.$$

□

## C.4 Details from MD (5) to PR (6)

Note that (5) is buyer-wise separable: for each $i$, we have (where $\frac{\partial}{\partial b_{ij}}\varphi_b(b) = \log \frac{p_j(b)}{v_{ij}}$ and $\mathcal{B}_i = B_i \cdot \Delta_{m+1}$)

$$(b_i^{t+1}, \delta_i^{t+1}) = \underset{(b_i, \delta_i)\in\mathcal{B}_i}{\arg\min} \sum_j \left( \log \frac{p_j(b^t)}{v_{ij}} - \log b_{ij}^t \right) b_{ij} - (\log \delta_i^t)\delta_i + \sum_j b_{ij}\log b_{ij} + \delta_i \log \delta_i$$

$$= \underset{(b_i, \delta_i)\in\mathcal{B}_i}{\arg\min} -\sum_j (\log b_{ij}^t)b_{ij} - (\log \delta_i^t)\delta_i + \sum_j b_{ij}\log b_{ij} + \delta_i \log \delta_i. \qquad (29)$$

By Lemma 6, for all $i, j$,

$$b_{ij}^{t+1} = B_i \cdot \frac{\frac{v_{ij}b_{ij}^t}{p_j(b^t)}}{\sum_\ell \frac{v_{i\ell}b_{i\ell}^t}{p_\ell(b^t)} + \delta_i^t}, \quad \delta_j^{t+1} = B_i \cdot \frac{\delta_i^t}{\sum_\ell \frac{v_{i\ell}b_{i\ell}^t}{p_\ell(b^t)} + \delta_i^t}. \qquad (30)$$

Let $p_j^t = p_j(b^t)$. Then, (30) can be written in terms of the allocations $x_{ij}^t = b_{ij}^t/p_j^t$ (which sum up to 1 over buyers $i$ for any item $j$) and leftover $\delta_i^t$, thus giving (6).

## C.5 Convergence of prices

Let $\eta^t = \max_j \frac{|p_j^t - p_j^*|}{p_j^*}$ be the relative price error, which can clearly be bounded by $\frac{\|p^t - p^*\|_1}{p_{\min}}$, where $p_{\min} = \min_j p_j > 0$ is given in Lemma 2. By Theorem 7 and strong convexity of KL divergence (w.r.t. $\|\cdot\|_1$), for $b^t$ and $p^t = p(b^t)$ generated by either PG or PR,

$$\frac{1}{2}\|p^t - p^*\|_1^2 \le D(p^t \| p^*) \le \varphi(b^t) - \varphi^*. \qquad (31)$$

Therefore, for PG, the quantities $\eta^t$, $\|p^t - p^*\|$ and $D(p^t \| p^*)$ all converge linearly to 0. For PR, they converge at $O(1/T)$.

We can further bound $\varphi(b^t) - \varphi^*$ by the duality gap. Specifically, given $b^t$, $p^t = p(b^t)$, let

$$b_i^t = \min\left\{ \min_j \frac{p_j^t}{v_{ij}}, 1 \right\}.$$

Then, $(p^t, \beta^t)$ is feasible to (22). By weak duality,

$$\varphi(b^t) - \varphi^* \le \varphi(b^t) + g(p^t, \beta^t), \qquad (32)$$

where $g(p, \beta)$ is the (minimization) objective of (22). Combining the above, we have

$$\eta^t \le \frac{\sqrt{2\left(\varphi(b^t) + g(p^t, \beta^t)\right)}}{p_{\min}}.$$

Note that the above holds for $b^t$ from either PG or PR. Although neat in theory, numerical experiments suggest that the above bound can be loose and is not suitable as a termination criteria.

 # D  Leontief utilities

 ## D.1  Derivation of (7)

 The primal EG (1) under Leontief utilities $u_i(x_i) = \min_{j \in J_i} \frac{x_{ij}}{a_{ij}}$ can be written in both $x$ and $u$:

$$\min_{u,\, x} \; -\sum_i B_i \log u_i$$
$$\text{s.t. } u_i \leq \frac{x_{ij}}{a_{ij}}, \; \forall j \in J_i, \; \forall i \in [n],$$
$$\sum_i x_{ij} \leq 1, \; \forall j \in [m],$$
$$x \geq 0, \; u \geq 0.$$

 Clearly, it can also be written in terms of $u_i$ only as follows:

$$\min -\sum_i B_i \log u_i \;\text{ s.t. } \sum_{i \in I_j} a_{ij} u_i \leq 1, \; \forall j, \; u \geq 0. \qquad (33)$$

 Let $p_j \geq 0$ be the dual variable associated with constraint $\sum_{i \in I_j} a_{ij} u_i \leq 1$. The Lagrangian is

$$\mathcal{L}(u, p) = -\sum_i B_i \log u_i + \sum_j p_j \left( \sum_{i \in I_j} a_{ij} u_i - 1 \right)$$
$$= -\sum_j p_j + \sum_i \left[ -B_i \log u_i + \langle a_i, p \rangle u_i \right].$$

 Note that minimizing $\mathcal{L}$ w.r.t. $u$ can be performed separably for each $u_i$. For any $i$ such that
 $\sum_{j \in J_i} p_j > 0$, by first-order stationarity condition, the term $-B_i \log u_i + \langle a_i, p \rangle u_i$ is minimized
 at $u_i^*(p) = \frac{B_i}{\langle a_i, p \rangle}$ with minimum value $B_i(1 - \log B_i) + B_i \log \langle a_i, p \rangle$. If $\sum_{j \in J_i} p_j = 0$, the term
 approaches $-\infty$ as $u_i \to \infty$. Therefore, the dual objective is

$$g(p) = \begin{cases} -\sum_j p_j + \sum_i B_i \log \langle a_i, p \rangle + \sum_i B_i (1 - \log B_i) & \text{if } p \geq 0 \text{ and } \sum_{j \in J_i} a_{ij} p_j > 0 \\ -\infty & \text{o.w.} \end{cases}$$

 Hence the (Lagrangian) dual problem is $\max_p g(p)$. Its minimization form, up to the constant
 $-\sum_i B_i(1 - \log B_i)$, is

$$\min \left[ \sum_j p_j - \sum_i B_i \log \langle a_i, p \rangle \right] \; \text{ s.t. } p \geq 0. \qquad (34)$$

 By Theorem 1, we have the following.

 - An optimal solution to (34) gives equilibrium prices.
 - A market equilibrium $(x^*, p^*)$ satisfies $\langle p^*, x_i \rangle = B_i$ for all $i$ and $\sum_i x_{ij}^* = 1$ for all $j$.
 Therefore, we have $\sum_j p_j^* = \|B\|_1$.

 Therefore, we can add the constraint $\sum_j p_j = \|B\|_1$ to (34) without affecting any optimal (equilib-
 rium) solution. This leads to (7).

 ## D.2  Proof of Lemma 3

 let $p$ be any feasible solution to (7). Since $\sum_j p_j = \|B\|_1$, we have $\langle a_i, p \rangle \leq \|a_i\|_\infty \|p\|_1 =$
 $\|a_i\|_\infty \|B\|_1$ for all $i$. Meanwhile, by Appendix D.1, at equilibrium, $p^*$ and primal variables $u_i^*$ satisfy
 $u_i^* = \frac{B_i}{\langle a_i, p^* \rangle}$ (by stationarity) and $u_i^* \leq$ utility of getting one unit of every item $= \min_{j \in J_i} \frac{1}{a_{ij}} =$
 $\frac{1}{\|a_i\|_\infty}$ for all $i$. Therefore $\langle a_i, p^* \rangle = \frac{B_i}{u_i^*} \leq \|a_i\|_\infty \|B\|_1$.

### D.3 Linear convergence of utilities

Note that the equilibrium utilities $u^*$ are clearly unique by (33). By the KKT stationary condition,

$$u_i^* = \frac{B_i}{\langle a_i, p^* \rangle}, \ \ \forall i$$

for equilibrium prices $p^*$. Therefore, an intuitive construction of $u^t$ is as follows. Let $p^t$ be the current iterate, $r_i^t = \langle a_i, p \rangle$. First compute $\tilde{u}_i^t = \frac{B_i}{r_i^t}$. Then, to satisfy the primal constraints $\sum_i u_i a_{ij} \leq 1$, take

$$u^t = \frac{\tilde{u}^t}{\max_j \sum_i u_i a_{ij}} = \frac{\tilde{u}^t}{\|a^\top \tilde{u}\|_\infty}.$$

Let $r^* = \langle a_i, p^* \rangle = \frac{B_i}{u_i^*}$ and $f^* = \arg\min_{p \in \mathcal{P}} \tilde{h}(ap) = \tilde{h}(r^*) = h(r^*)$. Strong convexity of $\tilde{h}$ implies $\frac{\mu}{2}\|r^t - r^*\|^2 \leq h(r^t) - f^*$. Furthermore, the mapping $r^t \mapsto \tilde{u}^t \mapsto u^t$ is Lipschitz continuous on $r^t \in [\underline{r}, \bar{r}]$. Therefore, $\|u^t - u^*\|$ converges to 0 linearly as well.

## E  Additional details on numerical experiments

For linear utilities, we generate market data $v = (v_{ij})$ where $v_{ij}$ are i.i.d. from standard Gaussian, uniform, exponential, or lognormal distribution. For each of the sizes $n = 50, 100, 150, 200$ (on the horizontal axis) and $m = 2n$, we generate 30 instances with unit budgets $B_i = 1$ and random budgets $B_i = 0.5 + \tilde{B}_i$ (where $\tilde{B}_i$ follows the same distribution as $v_{ij}$). See §6 for plots under random budgets and below for those under uniform budgets.

The termination conditions (on the vertical axis) are

$$\epsilon(p^t, p^*) \leq \eta, \ \eta = 10^{-2}, 10^{-3},$$

where $p^*$ is the optimal Lagrange multipliers of (1) computed by CVXPY+Mosek. Then, for $n = 100, 200, 300, 400$ and $n = 2m$, we repeat the above with termination conditions

$$\text{dgap}_t/n \leq \eta, \ \eta = 10^{-3}, 10^{-4}, 10^{-5}, 5 \times 10^{-6}.$$

For QL utilities, we repeat the above (same random $v$, same sizes and termination conditions) using budgets $B_i = 5(1 + \tilde{B}_i)$. This is to make buyers have nonzero bids and leftovers (i.e., $0 < \delta_i^* < B_i$) at equilibrium in most scenarios. In this case, $p^* = p(b^*)$, where $b^*$ is the optimal solution to (4) computed by CVXPY+Mosek. For QL, FW does not perform well in initial trials and is excluded in subsequent experiments.

For the linesearch subroutine $\mathcal{LS}_{\alpha,\beta,\Gamma}$ in PG (see Appendix A.5), we use parameters $\alpha = 1.02$, $\beta = 0.8$ and $\Gamma = 100L\|A\|^2$ throughout.

For Leontief utilities, in addition to $\text{dgap}_t/n \leq \eta$, we also use the termination condition $\epsilon(u^t, u^*) = \max_j \frac{|u_j^t - u_j^*|}{u_j^*} \leq \eta$, where $u^*$ is the optimal solution to EG under Leontief utilities (33) computed by CVXPY+Mosek.

**Computing the duality gap**  For linear utilities, the objective of the original Shmyrev's convex program (19) is

$$\varphi(b) = -\sum_{i,j}(\log v_{ij})b_{ij} + \sum_j p_j(b)\log p_j(b)$$

where $p_j(b) = \sum_i b_{ij}$. Recall the objective of the (EG) dual (20), equivalent to the dual of Shmyrev's (19),

$$g(p, \beta) = \sum_j p_j - \sum_i B_i \log \beta_i.$$

Given iterate $b^t$, let $p_j^t = p_j(b^t)$ and $\beta_i^t = \min_j \frac{p_j}{v_{ij}}$, which is finite since $v$ is nondegenerate and $p^t > 0$. The duality gap is computed via

$$\text{dgap}_t = \varphi(b^t) + g(p^t, \beta^t).$$

For QL utilities, it is computed similarly, that is, through (32). For Leontief utilities, it is computed using the construction in Appendix D.3.

Additional plots In §6, the plots for linear utilities are generated under random $B_i$. Here we present an augmented set of plots under different utilities, unit and random budgets $B_i$ and different termination conditions ($\mathrm{dgap}_t/n \le \eta$ or $\epsilon(p^t, p^*) \le \eta$). The legends are in the subplot [Linear utilities, dgap/$n \le$1e-3].

895

896

897

898

## Footnotes

[1]In fact, the bound $\log(mn)$ in [8, Lemma 13] (which assumes $\|B\|_1 = 1$) can be easily strengthened to $\log m$ via the above derivation. In other words, it does not depend explicitly on the number of buyers (but implicitly through $\|B\|_1$ in general).