[Reviews · NeurIPS 2020]

Review 1

Summary and Contributions: The paper considers algorithms for computing market equilibrium (i.e. who gets which items at what prices). It focuses on certain utility functions for the participants - linear, quasilinear, and Leontief. It shows that first-order convex optimization algorithms find market equilibria efficiently in these cases. Numerical experiments demonstrate practicality of the results and compare the different optimization algorithms.

Strengths: Well-written. Seems to carefully utilize extensive relevant literature and rigorously investigate the problem; is not just applying known methods (in my opinion) but has interesting contributions regarding the structure of this particular problem and these algorithms for it (e.g. discussion in lines 171-181). I think the results will be interesting and useful for future researchers and perhaps practitioners.

Weaknesses: The utility functions considered are somewhat limited. The paper is very compressed with all proofs in the appendix.

Correctness: I unfortunately was not able to check the appendix proof but did not see any red flags. The claims and justification make sense to me.

Clarity: Yes, I thought it was very good. In line 190, (mu,L)-s.c. is used before being defined; would be good to explain it or refer forward to the notation section.

Relation to Prior Work: Yes, I thought so.

Reproducibility: Yes

Additional Feedback: Nice job! Did you experimentally compare these methods to other algorithms (not in this paper) for market equilibrium? -------- After author response: Thanks for the detailed response. I remain positive about the paper.


Review 2

Summary and Contributions: This paper shows that maximizing something like utilities subject to some demand and allocation constraints can be written as convex optimization programs. The authors suggest some algorithms (pgd, mirror descent) for solving them.

Strengths: Some settings regarding market equilibrium are formulated as convex optimization problems. Theoretical guarantees in optimization aspect are provided. While I find the scenarios interesting, I have concerns regrading the significance and novelty of this paper (see below).

Weaknesses: It seems to be straightforward to write down some convex programs for the scenarios described in this paper. The technique is standard (e.g. Boyd and Vandenberghe). Once the convex formulations are written down, solving them by some convex optimization algorithms like PGD seems to be easy. The authors claim that this is one of the contributions (line 115 and 116). But, looking at Problems (1), (2), (4), (7), is anything challenging here? Or is there any non-convex formulations before this paper? Ref: Convex Optimization. Boyd and Vandenberghe Furthermore, Theorem 2,3,4,5 are all the standard/known convergence results of Projected Gradient Descent. What is new here? It seems to me that this paper just wrapped up some existing optimization results. In this sense, the contribution might not be very significant.

Correctness: yes

Clarity: yes

Relation to Prior Work: see *weakness*

Reproducibility: Yes

Additional Feedback: *** after rebuttal *** I've read the rebuttal and I upgrade my score accordingly. I shared similar concerns with another reviewer about the significance of the contributions, but most of the concerns have been clarified. I suggest the authors to rethink about the presentation (e.g. highlight the challenge and describe the formulations of prior works).


Review 3

Summary and Contributions: This paper adapts known first-order optimization methods to find the market equilibrium in the Fisher market model. They first show that when the utility function is linear, quasilinear and Leontief, the problem of interest is a convex program. In each of these settings, they provide convergence results and apply their method to synthetic datasets.

Strengths: The paper is original as it is in the intersection of two fields and uses convex optimization tools to solve a problem that arises in economics. It is perfectly relevant to the NeurIPS community and the theoretical statements are sound and make sense.

Weaknesses: After reading the authors response, I am convinced by the contribution of the authors. I really hope that they will include the answers that they provided to the final version of the paper because I find that it totally changes the reader's understanding of the paper. In particular, regarding the fact that obtaining a linear convergence rate is not obvious and was not done before. I re-evaluate my score. ============================================================== First, I found the novelty of the work a bit limited because not enough novelty was given in the "theory" side nor in the "numerical" side. The paper seems to "just" apply convex optimization methods to finding market equilibria and giving theoretical justifications for why this approach is sound. In particular, I feel that the novelty has not been highlighted enough and a question that can arise after a first lecture is how different this paper is compared to prior work that used convex optimization for computing market equlibria as [1]. While I imagine that the authors' contributions may be deeper and I would be happy to have clarifications on the main message of the paper. I believe that the contribution part should be more highlighted in the paper. Moreover, the choice of the first-order algorithms seem to be a bit arbitrary chosen. I imagine that this choice is motivated by the fact that those are the most standard methods for constrained optimization. However, It is not clearly explained why MD and FW would be more interesting than PGD for this problem. It is said in the paper that FW can be more interesting as it gives "sparse solutions" (l.173) but why is this important? The only motivation for MD seems that it allows to recover the PR dynamics. In terms of convergence rate, it seems that PGD is better than MD and FW but I imagine that FW and MD may have other type of advantages. Maybe the structure that is expected from a market equilibrium can help deciding which algorithm may be more suited but this is not clear in the paper. I think that better motivating the choice of these methods is important. Moreover, I think that as the related work part is not well structured and not exhaustive enough (see below), I felt that this contributed to weakening the contribution of the approach. Do you have examples of "large" markets where the gap between combinatorial and convex approaches may be huge? I think that a numerical implementation highlighting the gap would have been useful. [1] Birnbaum, B., Devanur, N. R., & Xiao, L. (2011, June). Distributed algorithms via gradient descent for fisher markets. In Proceedings of the 12th ACM conference on Electronic commerce (pp. 127-136).

Correctness: The claims seem overall correct. and pretty clear Most of the results have been formulated for PGD and some have been given for FW and MD. I although have questions and concerns regarding the theory: a) I would have appreciated more intuition and background on the Eisenberg-Gale convex program. b) In the case of MD, how do you manage to get a sublinear convergence rate for the last iterate? This seems very surprising to me. Indeed, it is known that for smooth functions (as the function in (4) ), MD has convergence for the average iterate and cannot have a test point convergence as "averaging allows uncontrollable jumps of the function values at some iterations". [1, p.921] c) In practice do you optimize the original function of the extrapolated smooth and strongly convex function \tilde{h}? [1] Nesterov, Y., & Shikhman, V. (2015). Quasi-monotone subgradient methods for nonsmooth convex minimization. Journal of Optimization Theory and Applications, 165(3), 917-940.

Clarity: The paper is clearly written but I would have appreciated a bit more background on the market model.

Relation to Prior Work: I found that this part is one of the weaknesses of the paper. The related work is scattered throughout the text and it is quite difficult to understand what prior work has been done for finding market equilibria. In particular, I think it would have been good to have a section Related work and to explain the limitations of prior work and to contrast the related work with the approach.

Reproducibility: Yes

Additional Feedback: In the numerical experiments, I didn't find the legend if I am not wrong. So it was hard to see to what curve corresponds which method. Also, I would have appreciated some comments on the curves regarding the different distributions. I think one interesting experiment is heavy-tailed distribution as the values vary a lot. What method does work the best? Do you have some intuition why? Another popular method in first-order optimization is Dual Averaging. It is known to behave better than MD in regularized problems in the convex setting. Contrary to MD, the gradients are not added with "vanishing weights".The authors are not working in a regularized setting but it can be interesting to experimentally try this method.


Review 4

Summary and Contributions: They consider the problem of computing Fisher or competitive equilibrium (i.e., a pair of allocation vector and price vector) in the Fisher market setting with divisible items using First-order methods. They consider the cases where the buyers have linear, quasilinear, and Leontief utility functions and focus on the convergent properties of the projected gradient (PG), Franke-Wolfe (FW), and Mirror Descent (MD) of finding competitive equilibrium under these settings via reformulating the original convex program for competitive equilibrium as an optimization problem that is amicable to first-order methods. The main results are - Convergent results for linear utility function using PG (and a new reformation of the convex program) - Convergent results for quasilinear utility function using PG and MD (and a new reformation of the convex program) - Convergent results for Leontief utility function using PG (and a new reformation of the convex program) Finally, they consider experiments to show the tradeoff between different first-order methods under different settings.

Strengths: I think the idea itself is interesting and the authors provide alternative methods for computing competitive equilibrium. I would like to see more about efficiency experiments compared to the standard methods (convex program solvers and other heuristics) for computing competitive equilibrium. The experimental setups in the paper do not make such comparison (i.e., the motivation for this work is about large-scale markets) so it would be nice to see some of that. I am also outside of the area, I cannot attest to the significance and novelty of the work and didn't check the proofs carefully. I wonder how much of the existing techniques use a similar idea in the current work (e.g., reformulation and providing convergence results of the methods under reformulation). I think it is relevant to the NeurIPS community (e.g., I found a previous paper on market equilibrium topic from the previous conference "Causal Strategic Inference in Networked Microfinance Economies")

Weaknesses: See above.

Correctness: See above.

Clarity: I was able to go through most of the paper, but the paper is a bit technical for my understanding. I think the authors should explain some of the concepts a bit more directly.

Relation to Prior Work: I didn't find the related work section explicitly stated. I think it would be good for the authors to discuss them more explicitly.

Reproducibility: Yes

Additional Feedback: Note: I am reviewing this top-down. Please look over my comments before addressing the reviews. I am also out of the area, but I am doing my best to understand and review this paper. Title: Should be Title Case? Abstract: "Market equilibrium ..." -> it would be good to talk about the typical market setting in one sentence "utility functions ..." -> which component of the markets are you referring to? "simple first-order methods ..." -> what are they? efficient? "large-scale markets ..." -> with many goods and buyers? "structured smooth convex function over a polyhedral set" -> do they need to be constructed explicitly? "... projected gradient ..." -> of what function? "linear convergence" -> depends on what parameters? "To do so ..." -> which are you referring to ? "we utilize recent linear convergence ..." -> from where? "further refine the relevant constants ... in general and for our specific setups" -> not exactly clear what do you mean by that " ... linesearch achieves linear convergence under the Proximal-PŁ condition" -> I don't know too much about the meaning of this "For quasilinear ..." -> which settings are the previous sentences refer to? "Mirror Descent, sublinear last-iterate convergence ..." -> unclear to me " approximate solution" -> equilibrium? under what utility function settings? 1 Introduction "Market equilibrium is a classical model ... " -> would be good to add some citations "an optimal bundle ..." -> in what sense? "In this paper, " why are you pointing to spliddit.org? "... fairly divided among" -> subject to what fairness criteria? "a market equilibrium is computed ..." -> based on? the values of the n buyers are not clear here "the fair division of the items " -> what fairness criteria does it satisfy? "CEEI has ..." -> the notion of the mechanism? how is it suggested? Maybe I am missing the connection between the Fisher market equilibrium and CEEI? You have been talking about CEEI "Internet ad auctions ..." -> is this a different setting that market? "(approximate) CEEI " -> how do you define approximate? "related max Nash welfare solution" -> is there a max there? "...via lotteries..." -> how? in what setting? Is there a reason why the domain of u_i is not in [0,1] instead of R_+ since you consider unit supply? Is it possible for the value in D_i(p) to be smaller than B_i in general? Essentially, the net utility is negative Could you comment on the market clearance condition for item p_j = 0, it seems that it can be allocated still but not necessarily all of them "(with degree 1)" -> why is it with degree 1? what does it mean for x_i \ge 0? each x_ij \ge 0? "This captures many widely used utilities ..." -> does it include quasilinear? Are you missing for each j in [m] in (1)? "Nash social welfare" -> as I recall, Nash social welfare is defined differently in a different context (either without B_i or with sqrt(n) in some social choice context) Theorem 1: What if there isn't one? Then there isn't a market equilibrium? How do you define a solution then? "Leontief utilities have the form ..." -> what is this J_i and a_ij in the min? "Another notable convex program that captures ... " -> it would be good to define the terms later if it is not here "large markets" -> large in what parameters? n and m? " iterative first-order methods ..." -> what is it? " number of of nonzeros " -> unclear where it is coming from? "bounded polyhedral set" -> are you going to define this more formally later? "(μ,L)-s.c" -> unclear what it is should it be h(xA) instead of h(Ax)? or A \in R^{r x d} instead? "initial iterate" -> how do you determine this? "the (unique) minimum objective" -> unique? shouldn't min value be unique? Projected gradient: is the prod over X, the bounded polyhedral set??? How does it work? Did you define the gradient of f? "set of vertices" -> extreme points? "differentiable convex function d " -> how would you define d? "classical Polyak-Łojasiewicz" -> not sure what this is "convex program (4) " -> hasn't been defined yet " sublinear last-iterate convergence" -> how is this different from without "last-iterate"? How does it compare to PG then? (4) is only for QL utilities right? What about linear? "Extensive numerical experiments ..." -> for which utility functions? Is there any theoretical results for FW? When do you need x_+? Is it necessary? "denotes the Euclidean projection" -> how is this defined? "Kullback–Leibler (KL) divergence" -> i think you might have called this differently earlier It would be good to expand the definitions in the notation a bit more formally 2. Linear convergence of gradient-based methods Not sure what it means for range(A)? In Theorem 2, what is L there? How hard is it to compute Hoffman constant? Does theorem 2 hold for any x^0? I am not quite familiar with the concept of "linesearch" -> it would be good to describe it a bit more "Initial trials suggest that it ... " -> EG (2)? Can you compare and contrast the bounds of Theorem 4 and Theorem 2-3; it seems that they are two different bounds here and wasn't clear to me which one is better than the others I think it would be good to say how section 2 is connected to the main story of the paper; it seems that the results in this section are more about the convergence of PG for program (2) when q = 0 and q != 0 3 Linear utilities "v does not contain any zero row or column" -> how valid is this assumption? What happens if it does? ".... can be replaced by equalities without affecting any optimal solution" -> why is this true? "Subsequently, EG with ... " -> why? "In facr, the same is easily" -> fix, are you referring to the cost only or the bound? 4 Quasi-linear utilities "the money has value outside the market" -> unclear meaning here? where do the bids correspond to in the reformulation? 6 Experiments 'i.i.d. from standard ..." -> what distribution parameters? "and m = 2n" -> is there why this is the case? did you see much change when you vary m? not exactly clear to me what the title of the plots is saying x-axis, y-axis, and the lines (I think I see the legend on the right-hand side)? Why isn't Mirror Descent implemented here? I am guessing the message here is that, for different termination conditions, other PGLS or PR performs better in the experiments ******************************* After rebuttal **************************** We would like to thank the authors for providing the responses. I have read your responses.

[Author Response · NeurIPS 2020]

**To all Reviewers**  Thanks you for the insightful reviews. Reviewers 2 and 3 question the novelty and claim that
we merely apply known results. Prior to our work, there were *no* first-order method (FOM) for computing market
equilibria that converge linearly. Previous methods acheving (partial) linear convergence either (i) involve layers of
oracle abstraction (with costly oracle calls) (Bei et al. - Ascending-Price Algorithms...) or (ii) only exhibits initial linear
convergence under restrictive "large-market" assumptions (Cole & Tao - Balancing the Robustness...). Thus, the fact
that we are able to achieve a linear rate with something as simple as proximal (projected) gradient (PG) is, in our view, a
*strength* rather than weakness. Linear convergence of PG is not obvious either: applying the standard theory naively to
Eisenberg-Gale only yields a $1/T$ rate, and even this already requires our Lemma 1 through 3 in order to get a Lipschitz
constant. To get our linear rate (Thm 5), it is necessary to first realize that strong convexity and Lipschitz gradients can
be proved in the EG utility space using properties of market equilibrium (nobody seems to have realized this crucial
property before; a priori one only gets strict convexity), and utilizing our new bounds on equilibrium utilities (Lemma
1). This makes the reformulated problem satisfy the PL inequality (Eq. (13)). Finally, minor adjustments to existing
theorems are needed to tackle our setting. Similar ideas are needed in the case of QL and Leontief utilities. For all
cases, our new market-specific bounds on the equilibrium quantities (Lemma 1-3) are crucial as they ensure explicit
Lipschitz constants on gradients. In our paper, we will also make sure to clarify that these steps are indeed necessary.
Note also that for Thm 2 & 3, it was necessary to refine the known convergence guarantees, as explained in the texts (l.
590 & 631 in their proofs). Finally, we propose a practical linesearch scheme (Algorithm 1) which we implement in
the experiments. We give convergence guarantees (Theorem 4 & 10) under the Proximal-PL condition beyond strong
convexity. These are entirely new.

**To Reviwer 1**  Thanks for the suggestions. Our paper covers all well-known utility functions that can be handled within
the Eisenberg-Gale framework (i.e. convex, continuous, nonnegative and homogeneous utility functions). See e.g. [9]
which lists the class of standard utilities. The only standard ones missing from our paper are CES and Cobb-Douglas,
which yield "easy" convex programs (see Appendix A.6). Other utilities do not always yield convex programs.

**To Reviewer 2**  [It seems straightforward to write down...] In fact, there is an entire literature on convex programming
formulations of equilibrium problems. Many papers have been written on the celebrated Eisenberg-Gale convex
program, which is a very non-obvious construction. See, e.g., the well-known book "Algorithmic Game Theory."
Furthermore, our contributions are clearly *not* formulation of any convex program or *directly* calling any FOM. [The
authors claim that this is one of the contributions...] As stated in the summary of contributions and reiterated above,
when solving the convex programs using FOMs, in order to achieve linear convergence, we (need to and did) establish
new bounds on various equilibrium quantities (Lemma 1-3). See "To all Reviewers" for other technical contributions
along this line. Furthermore, utilizing the structure of problem (4) for QL utilities, Theorem 7 gives a $1/T$ last-iterate
convergence rate for Mirror Descent (MD), a result much stronger than the general theory of MD. We also show that it
yields highly efficient updates which are also interpretable dynamics (l. 284-287). With these clarifications, we hope
that you can reevaluate our contributions.

**To Reviewer 3**  ["just" applying standard theory] Note that our new bounds on equilibrium quantities (Lemma 1-3)
are crucial in establishing linear rates (Thm 5-7). Several other new techniques are needed as explained in "To all
Reviewers." [choice of FOMs arbitrary] We choose them because (i) PG achieves a linear rate due to our theory, (ii)
FW solution at iteration $t$ has $nt$ nonzeros (useful when computing a low accuracy solution with very large $m$), (iii)
PR/MD exploits problem structure to get $1/T$ last-iterate convergence, gives desirable dynamics (284-287) and is
fast when computing low-accuracy solutions (l. 326). To the best of our knowledge, dual averaging does not have
particular theoretical advantages over standard settings. [MD rate on last iterate] Indeed, the convergence result for
MD/PR (Thm 7) is nonstandard and specific to the Fisher market setting. It is because the entropy DGF on the prices
aligns with the objective to give a form of relative Lipschitz continuity (Lemma 14 in Appendix C.3). [heavy-tailed
distribution] Thanks for the suggestion! We ran experiments on Cauchy valuations: PGLS slows down significantly
but PR is not affected (across many sizes and accuracy levels). We conjecture that the Hoffman constant, similar to
the matrix condition number $\sigma_{\max}/\sigma_{\min}$, degrades significantly upon heavy-tail $v$ (this seems extremely difficult to
formalize). In contrast, the bound (Thm 7) for MD/PR is independent of $v$. Will include in the final version. [In practice
... $\tilde{h}$?] Quadratic extrapolations are used in the experiments. We hope that you can reevaluate our contributions given
the clarifications.

**To Reviewer 4**  Many thanks for the detailed, helpful comments. We will take all into account but due to space
constraints we respond only to a few here. The reason we do not compare to convex program solvers is that we are
motivated by large-scale settings, which conic solvers do not scale to. For smaller problems, open-source solvers such
as ECOS are much slower than our methods, while the industry-grade Mosek solver (high-performance C code; our
code is in python) outperforms PGLS, but is slower than PR and FW when a low-accuracy solution is needed. We will
point this out. In Theorem 1, existence of ME is guaranteed by existence of an EG optimal solution (which is then
guaranteed by the Extreme Value Theorem; it is often simply assumed but we will clarify). We do implement Mirror
Descent: the PR dynamics (6) are MD applied to the convex program (4) for QL markets.

[Meta-Review · NeurIPS 2020]

There was initially some concern that this paper didn't have highly significant results since it appears to be the case that your work is reducing the problem of equilibrium computation to convex optimization. However, after some consideration, the reviewers were swayed by your rebuttal and agreed that the results were not trivial, the topic was interesting, and this appears to be an interesting application of optimization methods. We decided to accept your paper for NeurIPS this year, but we would strongly suggest that you incorporate some of the comments in your rebuttal into the main document, to better highlight the novelty. We strongly encourage you to take into account Reviewer 4's comments, who carefully went through the paper.